# FaPS: A General and Fast Training Method for Diffusion Models

Xianglu Wang [1]   Bangxian Han [2]   Hu Ding [3]

## Abstract

Diffusion models have achieved state-of-the-art performance in image generation tasks. However, training powerful diffusion models remains time-consuming, which limits their practical deployment. In this paper, we revisit the learning dynamics of diffusion models through the lens of *spectral bias*, a phenomenon in which deep neural networks prioritize learning low-frequency modes. Through an empirical analysis of diffusion training, we observe that diffusion models exhibit a **dual** spectral bias. First, over training iterations, they fit low-frequency components earlier than high-frequency details. Second, along the diffusion timesteps, early denoising steps mainly reconstruct coarse low-frequency content, while high-frequency details emerge in later steps. Motivated by this observation, we propose Frequency-aware Patch Selection (**FaPS**), a general and fast training method for diffusion models that can be applied to both UNet and DiT backbones. Specifically, FaPS introduces a *frequency-aware gating* that adaptively selects image patches based on their frequency information and focuses computation only on the selected patches. Since the selection decisions are discrete and thus non-differentiable, we model the gating as a stochastic policy network and optimize it end-to-end using a policy gradient method. Our experiments demonstrate that FaPS achieves up to $3\times$ faster training while maintaining comparable or superior generation quality, and improves the performance of diffusion models in limited-data settings.

## 1. Introduction

Diffusion models (Ho et al., 2020; Song et al., 2020; Karras et al., 2022) have recently emerged as a powerful framework for image synthesis and have achieved state-of-the-art performance on several benchmarks, outperforming generative adversarial networks (GANs) (Goodfellow et al., 2014) in many settings (Dhariwal & Nichol, 2021). Beyond image generation, diffusion models have also achieved remarkable success in a wide range of applications, including video generation (Xing et al., 2024; Huang et al., 2024), text-to-image generation (Ramesh et al., 2022; Saharia et al., 2022), semantic segmentation (Brempong et al., 2022; Xu et al., 2023), time-series forecasting (Liu et al., 2024), and robust learning (Nie et al., 2022; Wang et al., 2023b; 2026).

Despite their remarkable success, diffusion models remain computationally expensive, as they demand substantial training time (Wang et al., 2025; 2023a; Zheng et al., 2024; Yao et al., 2024) and large-scale datasets (Wang et al., 2023a; 2026). For instance, training a vanilla diffusion model (Ho et al., 2020) on the FFHQ dataset (Karras et al., 2019) (at a resolution of $64 \times 64$) using 8 NVIDIA V100 GPUs requires approximately 4 days (Karras et al., 2022). Moreover, the best-performing models, such as DALL·E-2 (Ramesh et al., 2022), Stable Diffusion (Rombach et al., 2022), and Imagen (Saharia et al., 2022), rely on billion-scale image–text datasets (e.g., LAION-5B (Schuhmann et al., 2022)). Therefore, accelerating the training of diffusion models and improving data efficiency has become an urgent need for the broader development of generative AI and related applications.

In recent years, a number of elegant methods have been proposed to reduce the training cost of diffusion models (Wu et al., 2023; Wang et al., 2025). One representative approach is referred to as "Patch-wise Diffusion (PD)" (Wang et al., 2023a; Gao et al., 2023a;b), which has shown promising performance in several generation scenarios. For instance, Wang et al. (2023a) proposes a conditional score function at the patch level, where both the patch location in the original image and the patch size are treated as conditions. By training on **randomly** selected patches instead of full images, it significantly reduces the computational burden per iteration. Another line of work focuses on "timesteps" (Wang et al., 2025), primarily using re-weighting (Choi et al., 2022; Hang

[1]School of Artificial Intelligence and Data Science, University of Science and Technology of China, Anhui, China [2]School of Mathematics, Shandong University, Shandong, China [3]School of Computer Science and Technology, University of Science and Technology of China, Anhui, China. Correspondence to: Hu Ding <huding@ustc.edu.cn>.

*Proceedings of the 43rd International Conference on Machine Learning*, Seoul, South Korea. PMLR 306, 2026. Copyright 2026 by the author(s).

et al., 2023) and re-sampling (Xu et al., 2024a) techniques. For example, Wang et al. (2025) propose an asymmetric sampling strategy that reduces the frequency of steps from the convergence area while increasing the sampling probability in other regions and yields a speedup. However, the training cost of these methods are still not usually affordable for many researchers, especially for ones in academia. For instance, training Patch Diffusion on the LSUN dataset still requires approximately 64 NVIDIA V100 GPU days. A complete introduction of existing efficient diffusion training methods is provided in Appendix A.

These approaches reduce training costs by reallocating computation across patches (Wang et al., 2023a; Gao et al., 2023a) or timesteps (Wang et al., 2025), but a key lever for further speedup is understanding *what* information diffusion models learn during diffusion training, so that computation is not wasted on redundant or already-learned content. To this end, we conduct an in-depth study of the **learning dynamics** of diffusion models, since what the model focuses on learning "useful information" largely determines both the convergence speed and the final generation performance. It is well established that the information in an image is carried by a set of *spatial frequencies*, i.e., a set of planar sinusoids with unique frequencies and directions (Basri et al., 2020). Therefore, understanding the learning dynamics of diffusion models naturally calls for an analysis from a frequency perspective. Specifically, we revisit diffusion model training through the lens of *spectral bias* (Rahaman et al., 2019), which refers to the tendency of deep neural networks to learn low-frequency components earlier than high-frequency ones and has been extensively studied in recent years (Xu et al., 2024b). More critically, Yang et al. (2023) empirically observe that generated samples evolve from coarse structures to fine details along the diffusion timesteps, which can be interpreted as a manifestation of spectral bias in the sampling process. In parallel, Wang & Pehlevan (2026) derive an *inverse-variance spectral law* that shows the spectral bias arises during the diffusion training process.

Building on the above insights, we conduct an empirical study to further illustrate that spectral bias indeed exists in diffusion models. As shown in Figure 1a, we first visualize both the intermediate generations and their corresponding 2D Fourier magnitude spectra (Khayatkhoei & Elgammal, 2022). We observe that low-frequency components emerge early in the sampling process, as evidenced by energy concentration near the spectrum center, while high-frequency details progressively appear at later timesteps, indicated by increasing energy in the outer spectral regions. We then examine the spectral bias along the training iterations. At each checkpoint, we perform principal component analysis (PCA) (Pearson, 1901) on the generated samples and obtain a set of orthogonal principal components indexed in

the standard PCA order (see Appendix D for details). For natural images, it has been shown that the first few components mainly capture smooth, low-frequency structures, whereas components with larger indices tend to correspond to progressively higher-frequency details (Wang & Pehlevan, 2026). Therefore, for each principal component, we record the training iteration at which it emerges and plot this iteration as a function of the component index (Figure 1b). The resulting curve shows that low-index (i.e., low-frequency) components are learned much earlier than high-index (i.e., high-frequency) ones, which suggests a clear spectral bias along the training iterations. Taken together, these observations illustrate that diffusion models exhibit a **dual** spectral bias: they preferentially learn low-frequency information both across training iterations and along diffusion timesteps. This dual behavior suggests that explicitly controlling which frequencies are learned at each stage could potentially benefit training. From this frequency perspective, an interesting question naturally arises:

*(Q) How can the "dual spectral bias" of diffusion models be leveraged to design a more efficient training strategy that reduces training time while preserving or even improving generation quality?*

## 1.1. Our Main Contributions

To address the above problem, a straightforward idea is to sample training images whose frequency content is currently most beneficial for learning, so that the model focuses its gradient updates on the selected images at each stage of training. However, evaluating the frequency content of all training samples at each iteration would require an extra pass over the entire dataset, which is prohibitively expensive. Instead, we are inspired by Patch-wise Diffusion (PD) (Wang et al., 2023a; Gao et al., 2023a;b), which operates on randomly selected image patches rather than full images. Building on this framework, we propose **FaPS**, an adaptive frequency-aware patch selection method that employs a carefully designed frequency-aware gating module to select the most informative image patches, thereby significantly accelerating training and improving data efficiency.

Nevertheless, efficiently implementing this idea within diffusion training is not straightforward, as it involves two major challenges. **(C1)** The usefulness of a patch from a frequency perspective is inherently dynamic. Early in training and at small diffusion timesteps, patches that mainly contain coarse, low-frequency structure are most helpful for learning, whereas later in training and at larger timesteps, patches dominated by fine, high-frequency details become more informative (see Figure 1). In other words, how useful a given patch is depends on both the current training iteration and the diffusion timestep, and can change even for

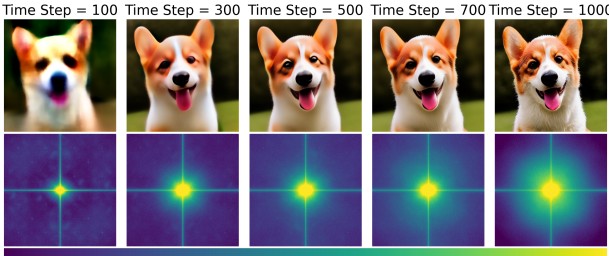

*(a)* Spectral evolution along diffusion timesteps. We visualize the centered 2D Fourier magnitude spectra of intermediate samples, where the horizontal and vertical axes correspond to spatial frequencies and brighter regions indicate larger magnitude.

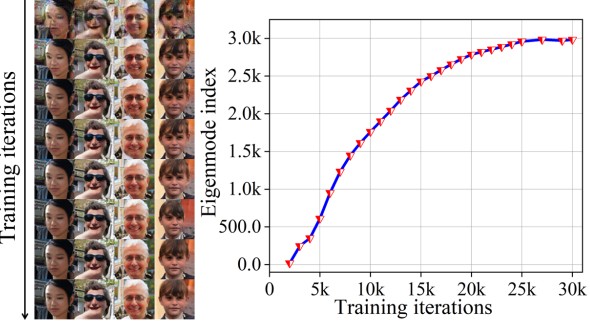

*(b)* Spectral bias across training iterations. *Left*: samples generated at different training iterations. *Right*: eigenmode index (vertical axis, indexing principal components of generated samples) versus training iteration, computed following (Wang & Pehlevan, 2026); dots mark the iteration at which each mode first emerges. Higher-index modes emerge progressively later, which indicate that low-frequency components are learned before high-frequency details.

*Figure 1.* Empirical evidence of dual spectral bias in diffusion models.

the same spatial location. This makes it difficult to predefine fixed rules for which patches should be selected. **(C2)** Patch selection itself is a discrete decision: at each step the algorithm must decide which patches to keep and which to skip. These discrete choices are non-differentiable and therefore cannot be optimized directly by standard backpropagation (Rumelhart et al., 1986), which introduces an additional obstacle to learning an effective patch selection policy.

To tackle these challenges, we first introduce an "adaptive" frequency-aware patch selection method. Specifically, we design a lightweight frequency-aware gating module that *learns* how to evaluate the importance of each patch using its frequency features and the current diffusion timestep, which enables the selection behavior itself to be trainable rather than fixed or heuristic. This trainable design allows the model to continuously update its assessment of patch importance as training progresses, naturally aligning the selection process with the dynamically evolving spectral patterns observed across iterations and diffusion timesteps. In addition, since the decision making process of the frequency-aware

gating module is discrete and thus non-differentiable, it cannot be optimized through standard backpropagation. To address this, we **view** the frequency-aware gating module as a *stochastic policy network* that outputs a probability distribution over patch selections, which can be optimized using a policy gradient method (Williams, 1992; Sutton et al., 1999). This formulation enables end-to-end training despite discrete decisions, thereby reducing redundant computation and accelerating training. Furthermore, we provide a convergence guarantee for FaPS, which shows that the joint optimization with the stochastic policy network remains theoretically well-behaved under standard assumptions.

We validate the effectiveness of our proposed FaPS through experiments on multiple datasets (e.g., FFHQ, LSUN, and AFHQv2) and across different model architectures (e.g., UNet (Karras et al., 2022) and DiT (Peebles & Xie, 2023)). The results demonstrate that FaPS achieves up to $3\times$ faster training while maintaining comparable or even superior generation quality. As another benefit of FaPS, we discover that it can improve the performance of diffusion models trained on limited-data settings.

## 2. Background

### 2.1. Diffusion-based Generative Models

In recent years, diffusion models (Ho et al., 2020; Peebles & Xie, 2023; Song et al., 2020; Karras et al., 2022) have emerged as a powerful class of deep generative models. A diffusion model consists of two processes: a *forward process* that gradually perturbs data into noise, and a *reverse process* that converts noise back to data. The forward process is typically predefined to transform any data distribution into a simple prior distribution (e.g., a standard Gaussian (Ho et al., 2020)), while the reverse process is learned by parameterizing transition kernels using deep neural networks (DNNs) (Yang et al., 2023). Formally, suppose we are given a dataset $\{\mathbf{x}_i\}_{i=1}^{N}$, where each sample is independently drawn from the data distribution $p_{\text{data}}(\mathbf{x})$. The goal is to construct a forward process $\{\mathbf{x}(t)\}_{t=0}^{T}$ indexed by a continuous time variable $t \in [0, T]$, such that the terminal state $\mathbf{x}(T) \sim p_T \approx \mathcal{N}(\mathbf{0}, \mathbf{I})$ serves as a tractable prior distribution. Following (Song et al., 2020), the forward diffusion process can be modeled as the solution to an stochastic differential equation (SDE):

$$d\mathbf{x} = \mathbf{f}(\mathbf{x}, t)\, dt + g(t)\, d\mathbf{w}, \tag{1}$$

where $\mathbf{f}(\cdot, t) : \mathbb{R}^d \to \mathbb{R}^d$ and $g(\cdot) : \mathbb{R} \to \mathbb{R}$ are diffusion and drift functions of the $\mathbf{x}(t)$, $\mathbf{w}$ denotes a standard Wiener process (a.k.a., Brownian motion), and $dt$ is an infinitesimal negative timestep. Then, for any diffusion process in the form of Eq. (1), Anderson (1982) shows that the reverse of it is also a diffusion process, running backwards in time and

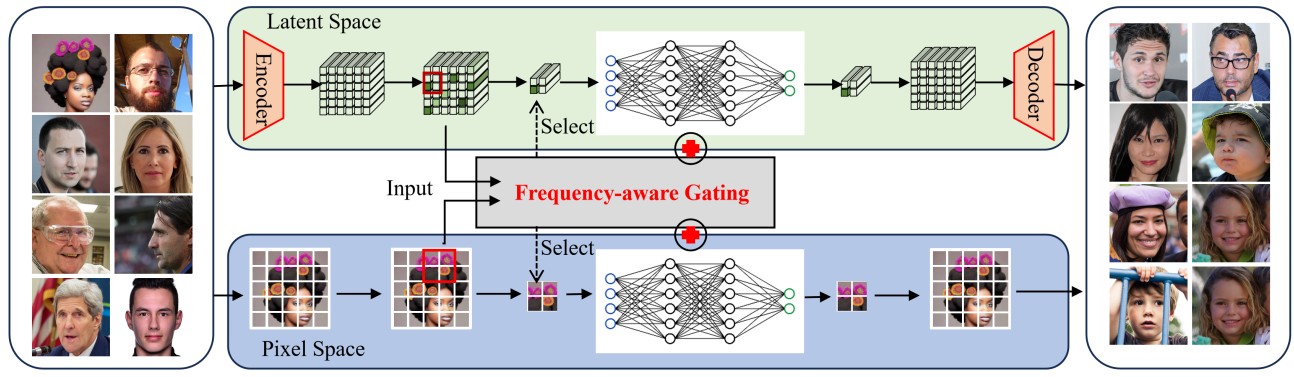

*Figure 2.* Overview of the proposed FaPS. The framework supports training in both latent space (*top*) and pixel space (*bottom*). Central to FaPS is a frequency-aware gating module that adaptively selects patches according to their frequency content and current timestep, which reduces training redundancy and accelerates convergence while preserving or even enhancing generation quality. The framework is compatible with different network backbones, such as UNet and DiT.

given by the reverse-time SDE (Song et al., 2020):

$$d\mathbf{x} = \left[\mathbf{f}(\mathbf{x}, t) - g(t)^2 \nabla_{\mathbf{x}} \log p_{\sigma_t}(\mathbf{x})\right] dt + g(t) d\bar{\mathbf{w}}, \quad (2)$$

where $\sigma_t$ is a positive noise scale at time $t$, and $\bar{\mathbf{w}}$ is a standard Wiener process when time flows backwards from $T$ to $0$. Intuitively, solutions to the reverse-time SDE (i.e., Eq. (2)) are diffusion processes that gradually convert noise to data. In addition, Song et al. (2020) establishes the *probability flow ODE*, an equivalent formulation whose trajectories share the same marginals as the reverse-time SDE, given by:

$$d\mathbf{x} = \left[\mathbf{f}(\mathbf{x}, t) - \frac{1}{2} g(t)^2 \nabla_{\mathbf{x}} \log p_{\sigma_t}(\mathbf{x})\right] dt, \quad (3)$$

where $\nabla_{\mathbf{x}} \log p_{\sigma_t}(\mathbf{x})$ is the score function (Hyvärinen & Dayan, 2005). Notably, the score function is the only unknown term in Eq. (3), and estimating it is therefore essential for simulating the reverse process. According to (Karras et al., 2022), a score-based model $\epsilon_\theta(\mathbf{x}, \sigma_t)$ is trained to approximate the true score function of the data distribution, i.e., $\nabla_{\mathbf{x}} \log p_{\sigma_t}(\mathbf{x})$. Specifically, Karras et al. (2022) design a denoiser $D_\theta(\mathbf{x}, \sigma_t)$ that is trained to minimize the expected $L_2$ denoising error for samples drawn from $p_{\text{data}}$, independently for each noise level $\sigma_t$. The corresponding training objective is given by

$$\mathbb{E}_{\mathbf{x} \sim p_{\text{data}}} \mathbb{E}_{\epsilon \sim \mathcal{N}(\mathbf{0}, \sigma_t^2 \mathbf{I})} \left\| D_\theta(\mathbf{x} + \epsilon; \sigma_t) - \mathbf{x} \right\|_2^2, \quad (4)$$

where $\epsilon$ is noise. The score function can be represented as

$$\mathbf{s}_\theta(\mathbf{x}, \sigma_t) = (D_\theta(\mathbf{x}; \sigma_t) - \mathbf{x})/\sigma_t^2. \quad (5)$$

### 2.2. Patch-wise Diffusion

Patch Diffusion (Wang et al., 2023a) is a UNet-based patch-wise training framework that reduces training cost and improves data efficiency. Instead of performing score matching on full images (see Eq. (5)), it learns the score function on random-sized patches. Specifically, for any $\mathbf{x} \sim p_{\text{data}}(\mathbf{x})$, Patch Diffusion **randomly** crops a set of patches $\{\mathbf{x}_k\}_{k=1}^K$ from $\mathbf{x}$, where $k = (i, j, s) \sim \mathcal{U}$. Here, $(i, j)$ denotes the top-left pixel coordinates and $s$ denotes the spatial size of the patch. The distribution $\mathcal{U}$ is uniform over the corresponding value ranges. Under this notation, the training loss of Patch Diffusion is formulated as

$$\mathcal{L}_{\text{patch}}(\theta) = \mathbb{E}_{\mathbf{x},t,\epsilon} \mathbb{E}_{k \sim \mathcal{U}} \| D_\theta(\mathbf{x}_k + \epsilon; \sigma_t, k) - \mathbf{x}_k \|_2^2. \quad (6)$$

In addition, the conditional score function Eq. (5) can be represented as

$$\mathbf{s}_\theta(\mathbf{x}, \sigma_t, k) = (D_\theta(\mathbf{x}_k; \sigma_t, k) - \mathbf{x}_k)/\sigma_t^2. \quad (7)$$

### 2.3. Fourier Spectrum Analysis

The Discrete Fourier Transform (DFT) (Bracewell, 1989; Brigham, 1988) is a fundamental mathematical tool that converts a signal from its original domain (e.g., time and space) into its frequency domain representation. In this work, we leverage the DFT to analyze the frequency characteristics of generated images. Specifically, given a single-channel image $\mathbf{x} \in \mathbb{R}^{H \times W}$ of spatial resolution $H \times W$, the DFT $\mathcal{F}[\cdot] : \mathbb{R}^{H \times W} \to \mathbb{C}^{H \times W}$ projects the image onto a basis of sine and cosine waves with different frequencies and phases, defined as

$$\mathcal{F}[\mathbf{x}](u, v) = \sum_{h=1}^{H} \sum_{w=1}^{W} \mathbf{x}(h, w) e^{-i2\pi\left(\frac{h}{H}u + \frac{w}{W}v\right)}, \quad (8)$$

where $\mathbf{x}(h, w)$ denotes the pixel value at spatial location $(h, w)$, $\mathcal{F}[\mathbf{x}](u, v)$ is the complex-valued Fourier coefficient corresponding to frequency $(u, v)$, $e$ is Euler's number, and $i^2 = -1$ denotes the imaginary unit.

# 3. Our Methodology

In this section, we propose FaPS, an adaptive method designed to enable faster and data-efficient training of diffusion models. In Section 3.1, we describe the architecture design of the frequency-aware gating module. In Section 3.2, we present the training objective of our method, derive the optimization algorithm of FaPS (Algorithm 1), and provide a convergence analysis of the proposed method. An overview of the overall framework is shown in Figure 2.

## 3.1. Frequency-aware Gating

As mentioned earlier (in Section 1.1), our method is based on the patch-wise training framework (Gao et al., 2023a; Wang et al., 2023a), which operates on image patches instead of full images. Under this formulation, patch selection directly determines the learning dynamics of the diffusion model. This observation motivates us to design a frequency-aware patch selection mechanism. Our core idea is that, if at each training step we choose patches whose frequency content matches the current stage of learning, the training objective concentrates the computational budget on the most informative patches, which in turn leads to faster training.

However, directly turning this frequency-based intuition into a practical patch selection rule is non-trivial. The usefulness of a patch is highly dynamic: it depends on both the diffusion timestep and the current iteration of training (as illustrated in Figure 1), so static heuristics based on fixed frequency bands or hand-crafted criteria quickly become suboptimal. To cope with this challenge, we introduce an adaptive frequency-aware gating module that takes the frequency representation of each patch, together with the diffusion timestep, as input and outputs a selection probability for each patch. In this way, the patch selection mechanism can automatically adjust its preference over low-frequency and high-frequency patches throughout training and align itself with the evolving dual spectral bias of the diffusion model. We next present the architecture of the proposed frequency-aware gating module.

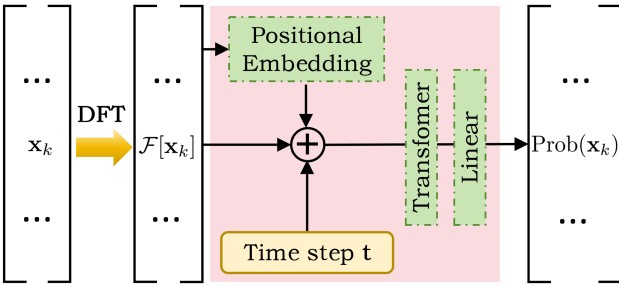

*Figure 3.* Illustration of the frequency-aware gating module.

**Architecture of frequency-aware gating.** Given an input image $\mathbf{x}$, we first divide it into $K$ patches $\{\mathbf{x}_1, \ldots, \mathbf{x}_K\}$.

For each patch $\mathbf{x}_k$, we apply a 2D DFT and obtain its frequency representation $\mathcal{F}[\mathbf{x}_k]$ (see Eq. (8)), which encodes the distribution of spatial frequencies within the patch. Recall that the relative importance of frequency components varies across diffusion timesteps (see Figure 1a); therefore the gating module needs to know the current diffusion timestep. Specifically, for the diffusion timestep $t$ sampled for this training iteration, we compute an embedding $\mathbf{e}_t$ and concatenate it with the frequency representation of every patch to form the patch token

$$\mathbf{h}_k = [\,\mathcal{F}[\mathbf{x}_k],\, \mathbf{e}_t\,]. \tag{9}$$

In addition, we add a learnable positional embedding to encode the spatial location of each patch. As illustrated in Figure 3, the frequency-aware gating module takes the sequence of patch tokens $\mathbf{h} := \{\mathbf{h}_1, \ldots, \mathbf{h}_K\}$ as input and produces a selection probability for each patch.

In practice, we implement the gating module as a lightweight Transformer encoder (Vaswani et al., 2017) with multi-head self-attention, followed by a shared linear projection that maps each token feature to a scalar score. This design allows the encoder to process the whole token sequence $\mathbf{h}$ jointly rather than scoring patches in isolation. Through self-attention, each patch token can attend to all others and aggregate their information; as a result, the representation of each patch is contextualized, capturing both its own content and its role within the image. To obtain selection probabilities, we apply a shared linear projection that produces a scalar logit for each patch. We then normalize these logits with a softmax function, which defines a categorical distribution $\{\mathrm{Prob}(\mathbf{x}_k)\}_{k=1}^K$ over patches. At each training iteration, we sample a single patch index $k$ and only forward the corresponding patch $\mathbf{x}_k$ through the diffusion backbone. In this way, the limited update budget is allocated in a data-dependent, learnable manner rather than determined by a fixed heuristic.

## 3.2. Optimization with Policy Gradient

Although frequency-aware gating outputs continuous selection probabilities for all patches, the act of choosing which patches to use in diffusion training is **discrete** and therefore non-differentiable. This prevents us from directly updating the parameters of the frequency-aware gating module using standard backpropagation (Rumelhart et al., 1986). To address this challenge, we view the frequency-aware gating module as a *stochastic policy* that samples patches conditioned on the patch tokens, and we optimize this policy using *policy gradient methods* (Williams, 1992; Sutton et al., 1999). In the following, we derive a policy gradient estimator for the frequency-aware gating parameters, which allows us to jointly train the diffusion model and the frequency-aware gating module.

Given a patch token sequence $\mathbf{h}$ (see Eq. (9)), frequency-aware gating parameterized by $\phi$ induces a stochastic sampling distribution $\pi_\phi(k|\mathbf{h})$ over patch indices; in practice we implement this as a categorical distribution (Murphy, 2012; Bishop et al., 1975) with class probabilities given by the selection probabilities $\mathrm{Prob}(\mathbf{x}_k)$. For a training example $\mathbf{x}$ and a sampled patch index $k$, we compute a patch diffusion loss $\ell_{\mathrm{patch}}(\theta; \mathbf{x}, t, \epsilon, k)$ (recall Eq. (6)) that only involves the selected patch. Our overall objective is then to minimize the expected diffusion loss under this stochastic selection process,

$$\mathcal{L}(\theta, \phi) = \mathbb{E}_{\mathbf{x},\epsilon,t}\, \mathbb{E}_{k \sim \pi_\phi(\cdot|\mathbf{h})} \big[ \ell_{\mathrm{patch}}(\theta; \mathbf{x}, t, \epsilon, k) \big], \quad (10)$$

where $\theta$ denotes the parameters of the diffusion model and $\phi$ denotes the parameters of the frequency-aware gating module.

We now derive the gradient of the objective in Eq. (10) with respect to $\phi$. By applying the *log-derivative trick* (De Smet et al., 2023), we obtain

$$\nabla_\phi \mathcal{L}(\theta, \phi) = \mathbb{E}_{\mathbf{x},\epsilon,t}\, \mathbb{E}_{k \sim \pi_\phi(\cdot|\mathbf{h})} \big[ \ell_{\mathrm{patch}}(\theta; \mathbf{x}, t, \epsilon, k) \\ \times \nabla_\phi \log \pi_\phi(k|\mathbf{h}) \big], \quad (11)$$

which is also known as the REINFORCE (Williams, 1992) gradient estimator. In practice, the estimator suffers from high variance, which may slow down or even destabilize training. Following standard practice in policy gradient methods (Sutton et al., 1999), we introduce a baseline "$b$" that does not depend on the sampled index $k$. Subtracting such a baseline does not change the expectation of the gradient, since $\mathbb{E}_{k \sim \pi_\phi(\cdot|\mathbf{h})}[b\, \nabla_\phi \log \pi_\phi(k|\mathbf{h})] = 0$, but it can reduce the variance of the estimator. This yields the variance-reduced gradient

$$\nabla_\phi \mathcal{L}(\theta, \phi) = \mathbb{E}_{\mathbf{x},\epsilon,t}\, \mathbb{E}_{k \sim \pi_\phi(\cdot|\mathbf{h})} \big[ (\ell_{\mathrm{patch}}(\theta; \mathbf{x}, t, \epsilon, k) - b) \\ \times \nabla_\phi \log \pi_\phi(k|\mathbf{h}) \big], \quad (12)$$

which remains unbiased while having lower variance. In our implementation, the baseline $b$ is given by an exponential moving average (EMA) of the patch loss with a decay factor of 0.99. The overall training procedure is outlined in Algorithm 1.

Following the theoretical analysis of stochastic gradient-based optimization algorithms (Bottou et al., 2018), we show that, under mild assumptions, the optimization procedure of FaPS enjoys a nonconvex convergence guarantee in terms of the expected gradient norm. Our main result is summarized in the following theorem.

**Theorem 1** (**Nonconvex convergence of FaPS**). *Suppose Assumptions 1 and 2 in Appendix B hold. Let $w_n = (\theta_n, \phi_n)$ be the model parameters generated by Algorithm 1 at iteration $n$, and define $\Delta = \mathcal{L}(w_1) - \min_w \mathcal{L}(w)$. If the step*

---

**Algorithm 1** FaPS

**Input:** Training dataset $\mathcal{D}$; Diffusion model parameters $\theta$; Patch selection policy parameters $\phi$; Patch extractor $\texttt{patchify}(\cdot)$; Initial baseline $b_0$; Learning rates $\eta_\theta$ and $\eta_\phi$; Number of timesteps $T$; A sequence of noise scales $\{\sigma_t\}_{t=1}^T$.

**Output:** Final model parameters $\theta$ and $\phi$.

1: Initialize $\phi, \theta, b \leftarrow b_0$
2: **for** $n = 1, \dots, N$ **do**
3:      Sample $\mathbf{x} \sim \mathcal{D}$, $t \sim \mathrm{Uniform}(\{1, \dots, T\})$
4:      */* Get patches and frequency features */*
5:      $\{\mathbf{x}_1, \dots, \mathbf{x}_K\} \leftarrow \texttt{patchify}(\mathbf{x})$
6:      $\mathbf{h} \leftarrow \{\}$
7:      **for** $k = 1, \dots, K$ **do**
8:          $\mathbf{h}_k = [\, \mathcal{F}[\mathbf{x}_k], \mathbf{e}_t \,]$
9:          */* Using DFT, i.e., Eq. (8) */*
10:     $\mathbf{h} = \mathbf{h} \cup \{\mathbf{h}_k\}$
11:      **end for**
12:      Sample $k \sim \pi_\phi(\cdot|\mathbf{h})$, $\epsilon \sim \mathcal{N}(\mathbf{0}, \sigma_t^2\mathbf{I})$
13:      */* Based on the loss (10), update the model $\theta$ */*
14:      $\theta_{n+1} \leftarrow \theta_n - \eta_\theta \nabla_\theta \ell_{\mathrm{patch}}(\theta_n; \mathbf{x}, t, \epsilon, k)$
15:      $b = \beta b + (1 - \beta)\ell_{\mathrm{patch}}(\theta_n; \mathbf{x}, t, \epsilon, k)$
16:      */* Based on the gradient (12), update the model $\phi$ */*
17:      $g_\phi(k|\mathbf{h}) := \nabla_\phi \log \pi_\phi(k|\mathbf{h})$
18:      $\phi_{n+1} \leftarrow \phi_n - \eta_\phi(\ell_{\mathrm{patch}}(\theta; \mathbf{x}, t, \epsilon, k) - b)g_\phi(k|\mathbf{h})$
19: **end for**

---

*sizes satisfy $\max\{\eta_\theta, \eta_\phi\} \leq 1/L$, then the iterates of Algorithm 1 satisfy*

$$\frac{1}{N}\sum_{n=1}^N \mathbb{E}\big[\|\nabla \mathcal{L}(w_n)\|_2^2\big] \leq \frac{2\Delta}{NL} + \frac{L^2(\sigma_\theta^2 + \sigma_\phi^2)}{4},$$

*where $L = (L_\theta + B_g L_0) + (G_\theta L_\pi + B_\ell L_g + B_\ell B_g L_\pi)$.*

The complete proof is given in Appendix B. Theorem 1 implies that incorporating our frequency-aware gating module does not affect the standard non-convex convergence behavior (Bottou et al., 2018): the average squared gradient norm still decreases on the order of $1/N$ as the number of iterations $N$ increases, and in the presence of stochastic gradient noise it converges to a neighborhood whose radius is bounded by $L^2(\sigma_\theta^2 + \sigma_\phi^2)/4$. In other words, FaPS preserves the standard convergence guarantee while introducing an adaptive, learnable patch selection mechanism.

## 4. Experiments

In this section, we first evaluate the generation quality and training speed of FaPS under different image resolutions and limited data settings, as detailed in Section 4.1. We then compare our approach with other state-of-the-art training acceleration methods on widely used benchmark datasets in Section 4.2.

**Experimental Setting.** We first adopt the UNet-based diffusion training framework EDM (Karras et al., 2022), which builds upon DDPM++ (Song et al., 2021) and ADM (Dhariwal & Nichol, 2021). Following Patch Diffusion (Wang et al., 2023a), we use the EDM-DDPM++ model to train on low-resolution datasets (e.g., $64 \times 64$), and employ an EDM-ADM coupling with Stable Diffusion (Rombach et al., 2022) to train on high-resolution datasets (e.g., $256 \times 256$). In addition, we also integrate our FaPS method into DiT-based diffusion models, specifically DiT-XL/2 (Peebles & Xie, 2023) and SiT-B/2 (Ma et al., 2024a). We calculate the Fréchet Inception Distance (FID) (Heusel et al., 2017) between $50,000$ generated images and all available real images. For the sampling process, we adopt Neural Function Evaluations (NFE) as a measure of sampling efficiency, where NFE denotes the number of forward evaluations of the neural network required to generate a single image. Additional experimental details are provided in Appendix C.

**Datasets.** For the low-resolution setting (i.e., $64 \times 64$), we select CelebA (Liu et al., 2015) and FFHQ (Karras et al., 2019). For the high-resolution setting (i.e., $256 \times 256$), we use the LSUN (Yu et al., 2015), MetFaces (Karras et al., 2020) and ImageNet (Deng et al., 2009). For the limited-data setting, we employ the AFHQv2 Cat, Dog, and Wild datasets (Choi et al., 2020).

### 4.1. Comparison with Baselines

In this section, we compare FaPS with baseline methods on several benchmark datasets and evaluate both generation quality (measured by FID) and training efficiency. Training efficiency is quantified by the "Speedup" ratio (Gao et al., 2023a;b; Qin et al.), defined as the training time of the baseline method divided by that of FaPS under the same experimental setting. Specifically, the baseline methods include DDPM (Ho et al., 2020), DDIM (Song et al., 2022), NCSN++ (Song et al., 2020), DDPM++ (Song et al., 2020), PNDM (Liu et al., 2022), EDM-DDPM++ (Karras et al., 2022), Soft Diffusion (Daras et al., 2022), ST (Kim et al., 2022), DeepCache (Ma et al., 2024b), Diff-Pruning (Fang et al., 2023), and Patch Diffusion (Wang et al., 2023a).

**Experiments on Low-resolution Dataset.** As shown in Table 1, FaPS reduces training time by about $3\times$ on both CelebA and FFHQ datasets compared with the EDM-DDPM++ baseline, while achieving comparable generation quality. For instance, FaPS attains nearly the same FID as EDM-DDPM++ (e.g., 1.65 vs. 1.66 on CelebA and 2.98 vs. 2.60 on FFHQ) with only about one third of the training time. In addition, Figure 4 shows the training curves of FID versus training time on these two datasets, where FaPS could converge much faster than the baseline.

**Experiments on High-resolution Dataset.** The results are summarized in Table 2, where FaPS improves both training

*Table 1.* FID ($\downarrow$) and Speedup ($\uparrow$) results on CelebA-$64 \times 64$ and FFHQ-$64 \times 64$ datasets. The best results are highlighted in **bold**, and the second-best results are underlined. Numerical results of the baseline methods are taken from their original publications. A dash ("–") denotes results not reported in the corresponding paper. "Speedup" denotes the ratio between the training time of the baseline method and that of FaPS under the same setting (larger is better).

| Methods | NFE | CelebA | FFHQ | Speedup |
|---|---|---|---|---|
| EDM-DDPM++ (Karras et al., 2022) | 50 | 1.66 | **2.60** | $1\times$ |
| DDPM (Ho et al., 2020) | 1000 | 3.51 | - | $1\times$ |
| DDIM (Song et al., 2022) | 100 | 6.53 | - | $1\times$ |
| NCSN++ (Song et al., 2020) | 1000 | 3.25 | - | $1\times$ |
| DDPM++ (Song et al., 2020) | 1000 | 1.90 | - | $1\times$ |
| PNDM (Liu et al., 2022) | 1000 | 2.71 | - | $1\times$ |
| Soft Diffusion (Daras et al., 2022) | 300 | 1.85 | - | $1\times$ |
| ST (Kim et al., 2022) | 1000 | 2.90 | - | $1\times$ |
| Patch Diffusion (Wang et al., 2023a) | 50 | 1.77 | 3.11 | $2\times$ |
| **FaPS (Ours)** | 50 | **1.65** | 2.98 | $\sim 3\times$ |

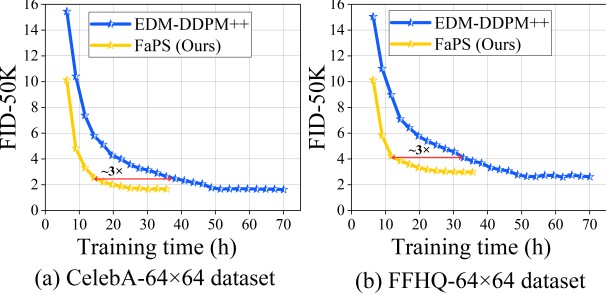

*Figure 4.* Training curves of FID versus training time on CelebA-$64 \times 64$ (left) and FFHQ-$64 \times 64$ (right) datasets.

speed and FID over latent diffusion baselines. Compared with LDM-ADM, our method achieves about $3.3\times$ speedup while reducing FID from $4.32$ to $2.72$ on Bedroom and from $4.66$ to $2.73$ on Church. Relative to the more recent LPDM-ADM, FaPS attains slightly better or comparable FID ($2.72$ vs. $2.75$ on Bedroom and $2.73$ vs. $2.66$ on Church) with a larger speedup ($3.3\times$ vs. $2.0\times$).

**Experiments on Limited-size Dataset.** It is well known that diffusion models require large amounts of data for stable training (Ho et al., 2020; Karras et al., 2022), which substantially limits their applicability in limited-data settings. Since FaPS trains on patches cropped at different spatial scales (Wang et al., 2023a), a single image can be expanded into hundreds of training patches (see Algorithm 1). This naturally raises the question of whether our approach should be regarded as a particular form of data augmentation that improves generation quality in the limited-data regime. To evaluate this hypothesis, we conduct experiments on the AFHQv2 dataset, which consists of three small subsets: Cat ($5,313$ images), Dog ($4,739$ images), and Wild ($4,738$ images). The results are summarized in Table 3, where FaPS achieves the lowest FID on all three AFHQv2 subsets while providing about $3.5\times$ speedup over EDM-DDPM++. Com-

*Table 2.* FID (↓) and Speedup (↑) results on LSUN-256 × 256 (i.e., Bedroom and Church) datasets. The best results are highlighted in **bold**, and the second-best results are underlined. Numerical results of the baseline methods are taken from their original publications. A dash ("-") denotes results not reported in the corresponding paper.

| Methods | NFE | Bedroom | Church | Speedup |
|---|---|---|---|---|
| LDM-ADM (Wang et al., 2023a) | 50 | 4.32 | 4.66 | 1× |
| DDPM (Ho et al., 2020) | 1000 | 4.90 | 7.89 | - |
| DDIM (Song et al., 2022) | 50 | 6.62 | 10.58 | - |
| Diff-Pruning (Fang et al., 2023) | 100 | 18.60 | 13.90 | - |
| DeepCache (N=2) (Ma et al., 2024b) | 1000 | 6.69 | 11.31 | - |
| LPDM-ADM (Wang et al., 2023a) | 50 | 2.75 | **2.66** | 2.0× |
| **FaPS (Ours)** | 50 | **2.72** | 2.73 | **3.3×** |

pared with Patch Diffusion, which also relies on patch-based training and can itself be viewed as a data augmentation baseline, FaPS further reduces FID (e.g., 3.11 vs. 2.98 on Cat, 4.80 vs. 4.73 on Dog, and 1.93 vs. 1.63 on Wild) under the same NFE, and increases the speedup from 2× to roughly 3.5×. These results demonstrate that FaPS can improve generation quality under limited training data. In addition, Figure 5 shows image samples generated by FaPS on the AFHQv2 subsets.

*Table 3.* FID (↓) and Speedup (↑) results on AFHQv2-64 × 64 (i.e., Cat, Dog, and Wild) datasets. The best results are highlighted in **bold**. Numerical results of the baseline methods are taken from their original publications.

| Data | Methods | FID | NFE | Speedup |
|---|---|---|---|---|
| **Cat** | EDM-DDPM++ (Karras et al., 2022) | 4.60 | 50 | 1× |
| | Patch Diffusion (Wang et al., 2023a) | 3.11 | 50 | 2× |
| | **FaPS (Ours)** | **2.98** | 50 | **3.6×** |
| **Dog** | EDM-DDPM++ (Karras et al., 2022) | 4.94 | 50 | 1× |
| | Patch Diffusion (Wang et al., 2023a) | 4.80 | 50 | 2× |
| | **FaPS (Ours)** | **4.73** | 50 | **3.4×** |
| **Wild** | EDM-DDPM++ (Karras et al., 2022) | 2.59 | 50 | 1× |
| | Patch Diffusion (Wang et al., 2023a) | 1.93 | 50 | 2× |
| | **FaPS (Ours)** | **1.63** | 50 | **3.6×** |

### 4.2. Comparisons with other acceleration methods

In this section, we compare FaPS with recent training acceleration methods on the DiT (Peebles & Xie, 2023; Ma et al., 2024a) architecture. Note that Patch Diffusion (Wang et al., 2023a) is a UNet-based patch-wise training framework and thus is not directly applicable to DiT. For the DiT experiments, we therefore adopt Masked Diffusion Transformer (MDT) (Gao et al., 2023a;b) as the patch-wise training counterpart of DiT and implement FaPS on top of this framework. We compare FaPS with representative timestep-level optimization baselines, including CLTS (Xu et al., 2024a), P2 (Choi et al., 2022), Min-SNR (Hang et al., 2023), and SpeeD (Wang et al., 2025). These methods mainly improve diffusion training by adjusting the contribution of different timesteps, whereas FaPS operates from a different perspective. All approaches are trained with

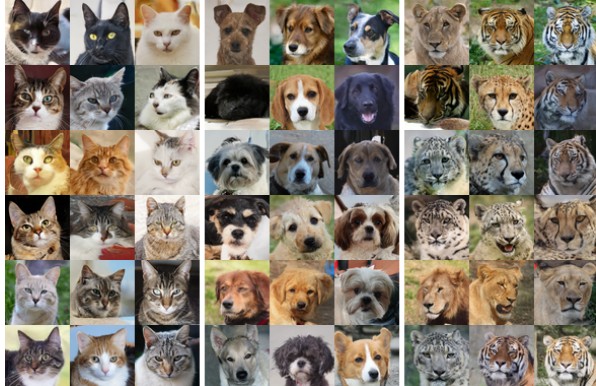

*Figure 5.* Samples generated by FaPS trained on the AFHQv2 Cat, Dog, and Wild subsets (from left to right).

DiT-XL/2 (Peebles & Xie, 2023), the XL model variant with patch size 2, for 100 GPU hours. The results are summarized in Table 4, which suggest that FaPS achieves the largest training speedup among all methods (3.8×) under the same training settings. Meanwhile, it maintains comparable generation quality, with FID of 21.88 on MetFaces and 10.44 on FFHQ, close to MDT. In addition, we combine FaPS with SpeeD (Wang et al., 2025) (a representative recent training acceleration method) in Table 4 to evaluate compatibility. The results show that FaPS provides an additional speedup of at least 1.4× on top of SpeeD, which indicates the good compatibility of FaPS. We provide further comparisons between FaPS and other methods that primarily operate on the latent autoencoder in Appendix C.2.

*Table 4.* FID results on MetFaces-256 × 256 and FFHQ-256 × 256. All methods are trained using DiT-XL/2 for 100 GPU hours on NVIDIA A6000 GPUs. The best results are indicated in **bold**.

| Methods | NFE | MetFaces | FFHQ | Speedup |
|---|---|---|---|---|
| DiT-XL/2 (Peebles & Xie, 2023) | 250 | 29.34 | 12.58 | 1.0× |
| MDT (Gao et al., 2023a) | 250 | **21.67** | 10.01 | 3.4× |
| CLTS (Xu et al., 2024a) | 250 | 23.52 | 12.66 | 1.5× |
| SpeeD (Wang et al., 2025) | 250 | 22.13 | **9.96** | 3.2× |
| P2 (Choi et al., 2022) | 250 | 22.67 | 16.32 | 1.4× |
| Min-SNR (Hang et al., 2023) | 250 | 28.61 | 13.14 | 1.4× |
| **FaPS (Ours)** | 250 | 21.88 | 10.44 | **3.8×** |
| SpeeD (Wang et al., 2025) | 250 | 22.13 | **9.96** | 1.0× |
| SpeeD + FaPS | 250 | 21.94 | 10.85 | **1.4×** |

## 5. Conclusion

In this paper, we revisited the learning dynamics of diffusion models through the lens of spectral bias and identified a dual spectral bias across training iterations and diffusion timesteps. Leveraging this insight, we proposed FaPS, an adaptive frequency-aware patch selection framework with a carefully designed gating module trained via policy gradients. The experiments show that FaPS can accelerate diffusion training while preserving or even improving generation quality, especially in limited-data settings.

## Limitations

Our theoretical analysis provides a standard constant step-size stochastic optimization bound on the averaged gradient norm. Since the bound contains a variance-dependent residual term, it should be viewed as an approximate-stationarity result rather than an exact convergence guarantee. We leave tighter analyses for future work.

## Acknowledgements

The authors would like to thank the anonymous reviewers for their valuable comments and suggestions. This work was partially supported by the National Key Research and Development Program of China (No. 2021YFA1000900), the National Natural Science Foundation of China (No. 62272432, No. 62432016), and the Natural Science Foundation of Anhui Province (No. 2208085MF163). The experiments and model training were supported by the robotic AI-Scientist platform of Chinese Academy of Science.

## Impact Statement

This paper presents work whose goal is to advance the field of Machine Learning. There are many potential societal consequences of our work, none which we feel must be specifically highlighted here.

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

## A. Related Work

**Diffusion Models.** Diffusion models (Ho et al., 2020; Peebles & Xie, 2023; Song et al., 2020; Karras et al., 2022) have become a leading paradigm for deep generative modeling. They achieve state-of-the-art performance on image synthesis tasks (Dhariwal & Nichol, 2021; Ho et al., 2020), often outperforming alternative generative frameworks, including variational autoencoders (VAEs) (Kingma et al., 2013), energy-based models (EBMs) (LeCun et al., 2006; Ngiam et al., 2011), and normalizing flows (NFs) (Rezende & Mohamed, 2015; Kobyzev et al., 2020). Compared with these approaches, diffusion models avoid posterior inference difficulties in VAEs, circumvent the instability of adversarial training in GANs, and do not require the architectural constraints typically imposed by NFs; moreover, EBMs often rely on expensive sampling procedures during learning. As a result, diffusion models have demonstrated strong potential across a range of domains, including computer vision (Baranchuk et al., 2022; Song et al., 2023), natural language processing (Yu et al., 2022; Li et al., 2022), and multi-modal modeling (Saharia et al., 2022; Zhu et al., 2023). Meanwhile, many methods have been proposed to further improve diffusion models, either by enhancing empirical performance (Nichol & Dhariwal, 2021; Song et al., 2021) or by extending their modeling capacity from a theoretical perspective (Lu et al., 2022b;a; Zhang & Chen, 2023).

**Acceleration in Diffusion Models.** A number of elegant diffusion methods have been proposed to accelerate diffusion training. Several recent works speed up diffusion training by reallocating computation across diffusion timesteps (Choi et al., 2022; Hang et al., 2023; Wang et al., 2025; Xu et al., 2024a). The key idea is to prioritize the timesteps that contribute most to learning, for example by reweighting the loss or by sampling timesteps non-uniformly, so that fewer updates are spent on less informative parts of the diffusion trajectory. For instance, CLTS (Xu et al., 2024a) proposes a curriculum learning based time step schedule, gradually tuning the sampling probability from uniform to Gaussian by interpolation for acceleration; Speed (Wang et al., 2025) proposes an asymmetric sampling strategy that reduces the frequency of steps from the convergence area while increasing the sampling probability in other regions and achieves $3\times$ speedup. Another representative approach is a patch-wise training framework (Wang et al., 2023a; Gao et al., 2023a;b). Patch-wise training frameworks train diffusion models on image patches instead of full images. Concretely, each training step crops or selects one or a few patches from an image, runs the diffusion objective on the selected patches, and updates the model accordingly. By reducing the spatial size of each training instance, patch-wise training lowers per-step computation and can improve data efficiency, while still allowing the model to learn both global structure and local details over the course of training. In addition, several other lines of work have explored diffusion training acceleration, such as network architecture (Ryu & Ye, 2022) and diffusion algorithm (Wu et al., 2023).

**Spectral Bias.** Deep neural networks (DNNs) (LeCun et al., 2015) often fit target functions from low to high frequencies during training; this implicit frequency preference is commonly referred to as *spectral bias* (Rahaman et al., 2019; Xu et al., 2024b; Yang et al., 2023). This phenomenon has been further validated on real datasets (Xu et al., 2019; Chen et al., 2021; Frank et al., 2020). For instance, Xu et al. (2019) propose two methods (projection and filtering) to show that spectral bias appears in the training process of DNNs on high-dimensional benchmark datasets such as MNIST and CIFAR-10. Spectral bias has also been observed in deep generative models. In GANs, prior studies (Khayatkhoei & Elgammal, 2022; Chandrasegaran et al., 2021; Durall et al., 2020; Tancik et al., 2020) report a tendency to prioritize fitting low-frequency components while under representing high-frequency components of the data distribution. Similarly, Yang et al. (2023) show that diffusion models capture different frequency components at different stages of the denoising process, with a bias toward the dominant frequencies of the data. Unlike these studies, which primarily analyze the existence of spectral bias, our work focuses on exploiting frequency learning dynamics to design an efficient training framework for diffusion models.

## B. Proof of Theorem 1

The proof of Theorem 1 follows the analysis framework of (Bottou et al., 2018). Before proving the theorem, we first introduce the assumptions and a technical lemma needed for the convergence analysis. For convenience, we define the aggregate random variable $z := (\mathbf{x}, t, \epsilon)$ and the state variable $s := (\mathbf{h}(\mathbf{x}), t)$. Let $\ell_k(\theta; z) := \ell_{\text{patch}}(\theta; \mathbf{x}, t, \epsilon, k)$ and $g_\phi(k|s) := \nabla_\phi \log \pi_\phi(k|s)$. At iteration $n$, Algorithm 1 samples $z_n = (\mathbf{x}_n, t_n, \epsilon_n)$ and $s_n = (h(\mathbf{x}_n), t_n)$, and forms the stochastic gradient estimators

$$\hat{g}_n^\theta := \nabla_\theta \ell_k(\theta; z_n), \quad \hat{g}_n^\phi := (\ell_k(\theta; z_n) - b_n) g_{\phi_n}(k_n|s_n).$$

**Assumption 1.** *Assume that $\ell_k(\theta; z)$, $\nabla_\theta \ell_k(\theta; z)$, $\pi_\phi(\cdot|s)$, and $g_\phi(\cdot|s)$ satisfy the following Lipschitz conditions:*

$$\sup_z \|\ell_k(\theta; z) - \ell_k(\theta'; z)\|_2 \le L_0 \|\theta - \theta'\|_2$$

$$\sup_z \|\nabla_\theta \ell_k(\theta; z) - \nabla_\theta \ell_k(\theta'; z)\|_2 \le L_\theta \|\theta - \theta'\|_2$$

$$\sup_s \|\pi_\phi(\cdot|s) - \pi_{\phi'}(\cdot|s)\|_2 \le L_\pi \|\phi - \phi'\|_2$$

$$\sup_s \|g_\phi(\cdot|s) - g_{\phi'}(\cdot|s)\|_2 \le L_g \|\phi - \phi'\|_2,$$

*where $L_0, L_\theta, L_\pi, L_g$ are positive constants. Moreover, suppose that $\ell_k(\theta; z)$, $\nabla_\theta \ell_k(\theta; z)$, and $g_\phi(\cdot|s)$ are uniformly bounded, with bounds $B_\ell$, $G_\theta$, and $B_g$, respectively.*

**Assumption 2.** *Assume that there exist constants $\sigma_\theta^2, \sigma_\phi^2 > 0$ such that for all $n$,*

$$\mathbb{E}\left[\|\hat{g}_n^\theta - \nabla_\theta \mathcal{L}(\theta_n, \phi_n)\|_2^2 | \theta_n, \phi_n\right] \le \sigma_\theta^2,$$

$$\mathbb{E}\left[\|\hat{g}_n^\phi - \nabla_\phi \mathcal{L}(\theta_n, \phi_n)\|_2^2 | \theta_n, \phi_n\right] \le \sigma_\phi^2.$$

**Remark 1.** *Assumptions 1 and 2 are common assumptions made in the analysis of stochastic gradient–based optimization algorithms (Bottou et al., 2018). Assumption 1 ensures that the gradient of $\mathcal{L}(\theta, \phi)$ does not change arbitrarily quickly with respect to the parameter vector. Although ReLU (Nair & Hinton, 2010) is non-differentiable, recent studies (Allen-Zhu et al., 2019; Cao & Gu, 2019; Du et al., 2019) have shown that the loss function of overparameterized deep neural networks is semi-smooth. This helps justify Assumption 1.*

**Lemma 1.** *Under Assumption 1, the joint objective function $\mathcal{L}(\theta, \phi)$ is L-smooth. That is, there exists a constant $L > 0$ such that for all $w = (\theta, \phi)$ and $w' = (\theta', \phi')$,*

$$\|\nabla \mathcal{L}(w) - \nabla \mathcal{L}(w')\| \le L\|w - w'\|.$$

*Proof.* We first establish Lipschitz continuity separately for $\nabla_\theta \mathcal{L}$ and $\nabla_\phi \mathcal{L}$, and then combine the bounds. From (10), differentiating with respect to $\theta$ (noting that $\pi_\phi$ does not depend on $\theta$) yields

$$\nabla_\theta \mathcal{L}(\theta, \phi) = \mathbb{E}_z \left[ \sum_k \pi_\phi(k|s) \, \nabla_\theta \ell_k(\theta; z) \right]. \tag{13}$$

For $\phi$, we use the standard log-derivative identity:

$$\nabla_\phi \pi_\phi(k|s) = \pi_\phi(k|s) \, \nabla_\phi \log \pi_\phi(k|s) = \pi_\phi(k|s) \, g_\phi(k|s).$$

Thus,

$$\nabla_\phi \mathcal{L}(\theta, \phi) = \mathbb{E}_z \left[ \sum_k \pi_\phi(k|s) \, \ell_k(\theta; z) \, g_\phi(k|s) \right]. \tag{14}$$

**Step 1: Lipschitz bound for $\nabla_\theta \mathcal{L}$.**

Consider any $(\theta, \phi)$ and $(\theta', \phi')$. Define

$$\Delta_\theta := \nabla_\theta \mathcal{L}(\theta, \phi) - \nabla_\theta \mathcal{L}(\theta', \phi').$$

Using (13), we have

$$\Delta_\theta = \mathbb{E}_z \left[ \sum_k \pi_\phi(k|s) \, \nabla_\theta \ell_k(\theta; z) - \sum_k \pi_{\phi'}(k|s) \, \nabla_\theta \ell_k(\theta'; z) \right]. \tag{15}$$

Add and subtract $\sum_k \pi_\phi(k|s) \, \nabla_\theta \ell_k(\theta'; z)$ inside the expectation gives

$$\Delta_\theta = \mathbb{E}_z \left[ \sum_k \pi_\phi(k|s) \big( \nabla_\theta \ell_k(\theta; z) - \nabla_\theta \ell_k(\theta'; z) \big) + \sum_k [\pi_\phi(k|s) - \pi_{\phi'}(k|s)] \, \nabla_\theta \ell_k(\theta'; z) \right].$$

Taking norms and applying the triangle inequality together with Jensen's inequality, we obtain

$$\|\Delta_\theta\| \leq \mathbb{E}_z \left\| \sum_k \pi_\phi(k|s) \big(\nabla_\theta \ell_k(\theta; z) - \nabla_\theta \ell_k(\theta'; z)\big) \right\| + \mathbb{E}_z \left\| \sum_k \left[\pi_\phi(k|s) - \pi_{\phi'}(k|s)\right] \nabla_\theta \ell_k(\theta'; z) \right\|. \tag{16}$$

For the first term, using Assumptions 1 and $\sum_k \pi_\phi(k|s) = 1$,

$$\left\| \sum_k \pi_\phi(k|s) \big(\nabla_\theta \ell_k(\theta; z) - \nabla_\theta \ell_k(\theta'; z)\big) \right\| \leq \sum_k \pi_\phi(k|s) L_\theta \|\theta - \theta'\| = L_\theta \|\theta - \theta'\|. \tag{17}$$

For the second term, using the Hölder's inequality,

$$\begin{aligned}
\left\| \sum_k \left[\pi_\phi(k|s) - \pi_{\phi'}(k|s)\right] \nabla_\theta \ell_k(\theta'; z) \right\| &\leq \sum_k \left| [\pi_\phi(k|s) - \pi_{\phi'}(k|s)] \right| \cdot \|\nabla_\theta \ell_k(\theta'; z)\| \\
&\leq G_\theta \|\pi_\phi(\cdot|s) - \pi_{\phi'}(\cdot|s)\| \\
&\leq G_\theta L_\pi \|\phi - \phi'\|.
\end{aligned} \tag{18}$$

Substituting (18) and (17) into (16), we obtain

$$\|\nabla_\theta \mathcal{L}(\theta, \phi) - \nabla_\theta \mathcal{L}(\theta', \phi')\| \leq L_\theta \|\theta - \theta'\| + G_\theta L_\pi \|\phi - \phi'\|. \tag{19}$$

**Step 2: Lipschitz bound for $\nabla_\phi \mathcal{L}$.**

Consider any $(\theta, \phi)$ and $(\theta', \phi')$. Define

$$\Delta_\phi := \nabla_\phi \mathcal{L}(\theta, \phi) - \nabla_\phi \mathcal{L}(\theta', \phi').$$

Using (14), we have

$$\begin{aligned}
\Delta_\phi &= \mathbb{E}_z \left[ \sum_k \pi_\phi(k|s) \, \ell_k(\theta; z) \, g_\phi(k|s) - \sum_k \pi_{\phi'}(k|s) \, \ell_k(\theta'; z) \, g_{\phi'}(k|s) \right] \\
&= \mathbb{E}_z \left[ \sum_k \underbrace{\pi_\phi(k|s) \, (\ell_k(\theta; z) - \ell_k(\theta'; z)) \, g_\phi(k|s)}_{\textbf{(I)}} \right] \\
&\quad + \mathbb{E}_z \left[ \sum_k \underbrace{\pi_\phi(k|s) \, \ell_k(\theta'; z) \, (g_\phi(k|s) - g_{\phi'}(k|s))}_{\textbf{(II)}} \right] \\
&\quad + \mathbb{E}_z \left[ \sum_k \underbrace{(\pi_\phi(k|s) - \pi_{\phi'}(k|s)) \, \ell_k(\theta'; z) \, g_{\phi'}(k|s)}_{\textbf{(III)}} \right].
\end{aligned}$$

For term **(I)**, using Assumption 1, we have

$$\|\textbf{(I)}\| \leq \pi_\phi(k|s) \, |\ell_k(\theta; z) - \ell_k(\theta'; z)| \, \|g_\phi(k|s)\| \leq \pi_\phi(k|s) \, L_0 \|\theta - \theta'\| B_g.$$

Summing over $k$ yields

$$\left\| \sum_k \textbf{(I)} \right\| \leq B_g L_0 \|\theta - \theta'\|. \tag{20}$$

Similarly, for terms **(II)** and **(III)**, we have

$$\left\| \sum_k \textbf{(II)} \right\| \leq B_\ell L_g \|\phi - \phi'\|. \tag{21}$$

$$\left\|\sum_k (\text{III})\right\| \leq B_\ell B_g L_\pi \|\phi - \phi'\|. \tag{22}$$

Combining (20)–(22) and taking expectation over $z$ (which does not affect the constants), we obtain

$$\|\nabla_\phi \mathcal{L}(\theta, \phi) - \nabla_\phi \mathcal{L}(\theta', \phi')\| \leq B_g L_0 \|\theta - \theta'\| + (B_\ell L_g + B_\ell B_g L_\pi) \|\phi - \phi'\|. \tag{23}$$

Let $w = (\theta, \phi)$ and $w' = (\theta', \phi')$. Using the inequality $\|(a, b)\| \leq \|a\| + \|b\|$, we have

$$\|\nabla \mathcal{L}(w) - \nabla \mathcal{L}(w')\| \leq \|\nabla_\theta \mathcal{L}(\theta, \phi) - \nabla_\theta \mathcal{L}(\theta', \phi')\| + \|\nabla_\phi \mathcal{L}(\theta, \phi) - \nabla_\phi \mathcal{L}(\theta', \phi')\|. \tag{24}$$

Substituting (19) and (23) into (24), we have

$$\|\nabla \mathcal{L}(w) - \nabla \mathcal{L}(w')\| \leq (L_\theta + B_g L_0) \|\theta - \theta'\| + (G_\theta L_\pi + B_\ell L_g + B_\ell B_g L_\pi) \|\phi - \phi'\|. \tag{25}$$

Finally, since $\|\theta - \theta'\| \leq \|w - w'\|$ and $\|\phi - \phi'\| \leq \|w - w'\|$, we obtain

$$\|\nabla \mathcal{L}(w) - \nabla \mathcal{L}(w')\| \leq L \|w - w'\|, \tag{26}$$

where $L = (L_\theta + B_g L_0) + (G_\theta L_\pi + B_\ell L_g + B_\ell B_g L_\pi)$.

This completes the proof. $\qquad\square$

Now we are ready to prove Theorem 1.

*Proof.* Let $w = (\theta, \phi)$ and use the Euclidean norm on the concatenated vector. Since $\mathcal{L}$ is $L$-smooth (Lemma 1), for any $u, v$, we have

$$\mathcal{L}(v) \leq \mathcal{L}(u) + \langle \nabla \mathcal{L}(u), v - u \rangle + \frac{L}{2} \|v - u\|_2^2. \tag{27}$$

Apply (27) with $u = w_n$ and $v = w_{n+1}$. Algorithm 1 updates

$$\theta_{n+1} = \theta_n - \eta_{\theta, n} \hat{g}_n^\theta, \qquad \phi_{n+1} = \phi_n - \eta_{\phi, n} \hat{g}_n^\phi,$$

so that

$$w_{n+1} - w_n = \left(-\eta_{\theta, n} \hat{g}_n^\theta, -\eta_{\phi, n} \hat{g}_n^\phi\right). \tag{28}$$

Substitute (28) into (27), we obtain

$$\langle \nabla \mathcal{L}(w_n), w_{n+1} - w_n \rangle = -\eta_{\theta, n} \langle \nabla_\theta \mathcal{L}(w_n), \hat{g}_n^\theta \rangle - \eta_{\phi, n} \langle \nabla_\phi \mathcal{L}(w_n), \hat{g}_n^\phi \rangle, \tag{29}$$

and

$$\|w_{n+1} - w_n\|_2^2 = \eta_{\theta, n}^2 \|\hat{g}_n^\theta\|_2^2 + \eta_{\phi, n}^2 \|\hat{g}_n^\phi\|_2^2. \tag{30}$$

Therefore, the smoothness inequality yields the one-step bound

$$\mathcal{L}(w_{n+1}) \leq \mathcal{L}(w_n) - \eta_{\theta, n} \langle \nabla_\theta \mathcal{L}(w_n), \hat{g}_n^\theta \rangle - \eta_{\phi, n} \langle \nabla_\phi \mathcal{L}(w_n), \hat{g}_n^\phi \rangle + \frac{L}{2} \left(\eta_{\theta, n}^2 \|\hat{g}_n^\theta\|_2^2 + \eta_{\phi, n}^2 \|\hat{g}_n^\phi\|_2^2\right). \tag{31}$$

Since the stochastic gradient estimator is unbiased, we have

$$\mathbb{E}\left[\hat{g}_n^\theta | w_n\right] = \nabla_\theta \mathcal{L}(w_n), \qquad \mathbb{E}\left[\hat{g}_n^\phi | w_n\right] = \nabla_\phi \mathcal{L}(w_n). \tag{32}$$

Thus,

$$\mathbb{E}\left[\langle \nabla_\theta \mathcal{L}(w_n), \hat{g}_n^\theta \rangle | w_n\right] = \langle \nabla_\theta \mathcal{L}(w_n), \mathbb{E}\left[\hat{g}_n^\theta | w_n\right] \rangle \\ = \|\nabla_\theta \mathcal{L}(w_n)\|_2^2, \tag{33}$$

and similarly,

$$\mathbb{E}\left[\langle \nabla_\phi \mathcal{L}(w_n), \hat{g}_n^\phi \rangle | w_n\right] = \|\nabla_\phi \mathcal{L}(w_n)\|_2^2. \tag{34}$$

Plugging (33) and (34) into the conditional expectation of (31) yields

$$
\begin{aligned}
\mathbb{E}[\mathcal{L}(w_{n+1})|w_n] \leq{}& \mathcal{L}(w_n) - \eta_{\theta,n}\|\nabla_\theta\mathcal{L}(w_n)\|_2^2 - \eta_{\phi,n}\|\nabla_\phi\mathcal{L}(w_n)\|_2^2 \\
&+ \frac{L}{2}\Big(\eta_{\theta,n}^2\mathbb{E}\big[\|\hat{g}_n^\theta\|_2^2|w_n\big] + \eta_{\phi,n}^2\mathbb{E}\big[\|\hat{g}_n^\phi\|_2^2|w_n\big]\Big).
\end{aligned}
\tag{35}
$$

Under Assumption 2, we bound the second moments as follows:

$$
\begin{aligned}
\mathbb{E}\big[\|\hat{g}_n^\theta\|_2^2|w_n\big] &= \mathbb{E}\big[\|\nabla_\theta\mathcal{L}(w_n) + (\hat{g}_n^\theta - \nabla_\theta\mathcal{L}(w_n))\|_2^2|w_n\big] \\
&= \|\nabla_\theta\mathcal{L}(w_n)\|_2^2 + 2\big\langle\nabla_\theta\mathcal{L}(w_n),\mathbb{E}\big[\hat{g}_n^\theta - \nabla_\theta\mathcal{L}(w_n)|w_n\big]\big\rangle + \mathbb{E}\big[\|\hat{g}_n^\theta - \nabla_\theta\mathcal{L}(w_n)\|_2^2|w_n\big] \\
&\leq \|\nabla_\theta\mathcal{L}(w_n)\|_2^2 + \sigma_\theta^2.
\end{aligned}
\tag{36}
$$

Similarly,

$$
\mathbb{E}\big[\|\hat{g}_n^\phi\|_2^2|w_n\big] \leq \|\nabla_\phi\mathcal{L}(w_n)\|_2^2 + \sigma_\phi^2.
\tag{37}
$$

Substituting (36)–(37) into (35), we obtain

$$
\begin{aligned}
\mathbb{E}[\mathcal{L}(w_{n+1})|w_n] \leq{}& \mathcal{L}(w_n) - \eta_{\theta,n}\|\nabla_\theta\mathcal{L}(w_n)\|_2^2 - \eta_{\phi,n}\|\nabla_\phi\mathcal{L}(w_n)\|_2^2 \\
&+ \frac{L}{2}\Big(\eta_{\theta,n}^2\big(\|\nabla_\theta\mathcal{L}(w_n)\|_2^2 + \sigma_\theta^2\big) + \eta_{\phi,n}^2\big(\|\nabla_\phi\mathcal{L}(w_n)\|_2^2 + \sigma_\phi^2\big)\Big) \\
={}& \mathcal{L}(w_n) - \Big(\eta_{\theta,n} - \frac{L}{2}\eta_{\theta,n}^2\Big)\|\nabla_\theta\mathcal{L}(w_n)\|_2^2 \\
&- \Big(\eta_{\phi,n} - \frac{L}{2}\eta_{\phi,n}^2\Big)\|\nabla_\phi\mathcal{L}(w_n)\|_2^2 + \frac{L}{2}\big(\eta_{\theta,n}^2\sigma_\theta^2 + \eta_{\phi,n}^2\sigma_\phi^2\big).
\end{aligned}
\tag{38}
$$

Using the step size condition $\max\{\eta_{\theta,n},\eta_{\phi,n}\} \leq 1/L$, we have

$$
\eta_{\theta,n} - \frac{L}{2}\eta_{\theta,n}^2 \geq \frac{1}{2}\eta_{\theta,n}, \qquad \eta_{\phi,n} - \frac{L}{2}\eta_{\phi,n}^2 \geq \frac{1}{2}\eta_{\phi,n}.
\tag{39}
$$

Therefore, equation (38) can be rewritten as:

$$
\mathbb{E}[\mathcal{L}(w_{n+1})|w_n] \leq \mathcal{L}(w_n) - \frac{\eta_{\theta,n}}{2}\|\nabla_\theta\mathcal{L}(w_n)\|_2^2 - \frac{\eta_{\phi,n}}{2}\|\nabla_\phi\mathcal{L}(w_n)\|_2^2 + \frac{L}{2}\big(\eta_{\theta,n}^2\sigma_\theta^2 + \eta_{\phi,n}^2\sigma_\phi^2\big).
\tag{40}
$$

Taking full expectation of (40) and summing from $n = 1$ to $N$ yields

$$
\begin{aligned}
\mathbb{E}[\mathcal{L}(w_{N+1})] \leq{}& \mathcal{L}(w_1) - \frac{1}{2}\sum_{n=1}^N \eta_{\theta,n}\,\mathbb{E}\big[\|\nabla_\theta\mathcal{L}(w_n)\|_2^2\big] - \frac{1}{2}\sum_{n=1}^N \eta_{\phi,n}\,\mathbb{E}\big[\|\nabla_\phi\mathcal{L}(w_n)\|_2^2\big] \\
&+ \frac{L}{2}\sum_{n=1}^N\big(\eta_{\theta,n}^2\sigma_\theta^2 + \eta_{\phi,n}^2\sigma_\phi^2\big).
\end{aligned}
\tag{41}
$$

Rearranging terms gives

$$
\sum_{n=1}^N\big(\eta_{\theta,n} + \eta_{\phi,n}\big)\mathbb{E}\big[\|\nabla\mathcal{L}(w_n)\|_2^2\big] \leq 2\big(\mathcal{L}(w_1) - \mathbb{E}[\mathcal{L}(w_{N+1})]\big) + L\left(\sigma_\theta^2\sum_{n=1}^N \eta_{\theta,n}^2 + \sigma_\phi^2\sum_{n=1}^N \eta_{\phi,n}^2\right).
\tag{42}
$$

Recall that $\max\{\eta_{\theta,n},\eta_{\phi,n}\} \leq 1/L$ where $L = \big(L_\theta + B_g L_0\big) + \big(G_\theta L_\pi + B_\ell L_g + B_\ell B_g L_\pi\big)$, we can show that

$$
\frac{1}{N}\sum_{n=1}^N \mathbb{E}\big[\|\nabla\mathcal{L}(w_n)\|_2^2\big] \leq \frac{2\Delta}{NL} + \frac{L^2(\sigma_\theta^2 + \sigma_\phi^2)}{4},
$$

which completes the proof. $\qquad\square$

# C. More Experiment Results

In this section, we introduce the experimental setup in detail and present additional results.

## C.1. Details for Experimental Setup

We use U-Net (Karras et al., 2022) and DiT (Peebles & Xie, 2023) as the backbone architectures for our FaPS. For high-resolution experiments, we adopt the pre-trained VAE encoder and decoder from Stable Diffusion (Rombach et al., 2022) to map images into a lower-dimensional latent space, and perform training in this latent space. We use DiT-XL/2 as the diffusion backbone and optimize with AdamW using a constant learning rate of 1e-4 and no weight decay. Models are trained for 400K iterations on the MS-COCO training dataset and evaluated on the MS-COCO validation dataset using FID and CLIP score. We additionally adopt classifier-free guidance with a guidance scale of 1.5 for conditional sampling.

Following (Lin et al., 2019; Wang et al., 2023a), we normalize the coordinate system (either in pixel or latent space) into the range $[-1, 1]$ relative to the corresponding resolution. The upper-left corner is mapped to $(-1, -1)$ and the bottom-right corner to $(1, 1)$, as illustrated in Figure 6. Specifically, given an image of resolution $R \times R$ and a patch size $s$, the candidate pool consists of $(R - s + 1)^2$ possible patches. During training, however, we do not rely on all candidates; instead, the frequency-aware gating module (Section 3.1) adaptively selects the most informative ones, thereby avoiding redundant computation.

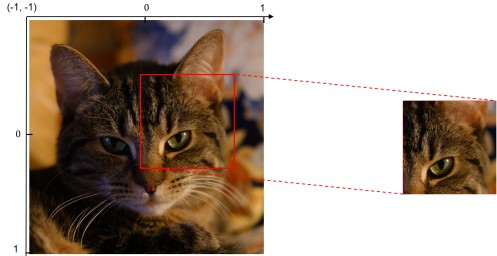

*Figure 6.* Illustration of patch location. An input image with normalized coordinates is shown, from which a patch is cropped at a specific spatial position (highlighted by the red box).

## C.2. Additional Results

**Runtime Overhead Analysis.** To further quantify the computational cost introduced by FaPS, we report the per-iteration overhead of the DFT-based feature extraction and the gating module in Table 5. The timing is measured under the same training configuration as our main experiments. Compared with a standard U-Net training iteration, the DFT feature extraction introduces only about 10 ms overhead, while the gating module introduces about 110 ms overhead. In total, the additional cost of FaPS is approximately 120 ms per iteration, corresponding to about 1.2% of the baseline iteration time. This indicates that the overhead introduced by FaPS is minor compared with the overall training computation.

*Table 5.* Per-iteration runtime breakdown. We report the additional overhead introduced by the DFT-based feature extraction and the gating module in FaPS. The overhead is measured relative to the standard U-Net training iteration.

| Computational Component | Time per Iteration | Relative Cost |
|---|---|---|
| Standard U-Net training iteration | 9760 ms | 100% |
| FaPS: DFT feature extraction | 10 ms | 0.1% |
| FaPS: gating module | 110 ms | 1.1% |
| Total FaPS overhead | 120 ms | 1.2% |

**Patch Selection Dynamics.** To analyze the behavior of the gating module during training, we examine the frequency composition of the selected patches. Specifically, for each sampled batch, we compute the frequency score of each patch using the DFT-based features and split all patches equally into high-frequency (HF) and low-frequency (LF) groups. Since the two groups have the same size, random selection would yield an expected HF proportion of 50%. Table 6 reports the

proportion of HF patches selected by FaPS at different timesteps and training stages. The results show that FaPS selects more HF patches as training progresses, and the HF proportion is also higher at larger timesteps. This suggests that the gating module learns non-trivial patch selection dynamics and provides empirical support for our spectral-bias motivation.

*Table 6.* Patch selection dynamics of the gating module. We report the proportion of high-frequency (HF) patches in the selected subset. Since HF and LF patches are split equally, random selection corresponds to 50%.

| Timestep | Early stage (1/3) | Later stage (2/3) |
|---|---|---|
| 300 | 42% | 55% |
| 700 | 52% | 67% |
| Random selection | 50% | 50% |

**Comparison with REPA-E.** Following recent works on accelerating diffusion training, we further compare FaPS with REPA-E (Leng et al., 2025) on the SiT-B/2 backbone (Ma et al., 2024a). The results are summarized in Table 7. All experiments are conducted on ImageNet under the same training setting. FaPS reduces the wall-clock training time per fixed iteration budget, achieving 400K iterations in about 25 hours compared with about 43 hours for the standard SiT-B/2 baseline. When trained for the same wall-clock budget, FaPS reaches competitive FID, indicating improved training efficiency. Moreover, combining FaPS with REPA-E further improves the FID from 23.21 to 21.07 under the same wall-clock budget, suggesting that FaPS is complementary to representation-alignment based acceleration methods.

*Table 7.* Comparison with recent diffusion training acceleration methods on the SiT-B/2 backbone.

| Method | Iterations | Training Time (h) | FID | Acceleration |
|---|---|---|---|---|
| SiT-B/2 (Ma et al., 2024a) | 400K | $\sim 43$ | 34.52 | $1.0\times$ |
| REPA-E (Leng et al., 2025) | 400K | $\sim 43$ | 23.21 | $\sim 2.1\times$ |
| FaPS | 400K | $\sim 25$ | 35.63 | – |
| FaPS | 690K | $\sim 43$ | 24.59 | $\sim 1.9\times$ |
| REPA-E + FaPS | 690K | $\sim 43$ | **21.07** | $\sim 2.5\times$ |

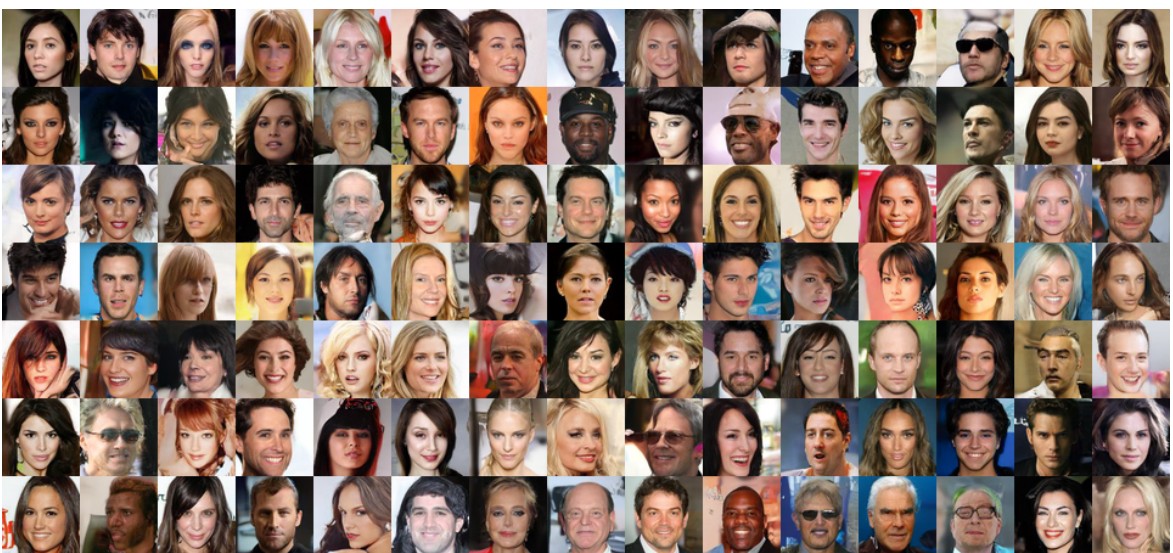

*Figure 7.* Randomly generated images from our method trained on CelebA-64×64. FID=1.65

# D. Empirical Study of Spectral Bias over Training

Following (Wang & Pehlevan, 2026), we empirically investigate spectral bias by monitoring when the PCA (Pearson, 1901) modes computed from the training data first emerge in the generated samples during training. Specifically, we utilize the FFHQ (Karras et al., 2019) dataset at $32 \times 32$ resolution[1]. For each image, we first flatten it into a $d$-dimensional vector ($d = 3072$) and then stack these vectors row-wise into a data matrix $\mathbf{X} \in \mathbb{R}^{N \times d}$, where $N$ ($N \gg d$) denotes the number of training images. In addition, we normalize these pixel intensities from $[0, 1]$ to $[-1, 1]$ via $\mathbf{x} \mapsto \frac{\mathbf{x} - 0.5}{0.5}$. To obtain the principal components of the real training data for comparison, we first compute the empirical covariance $\Sigma = \mathrm{Cov}(\mathbf{X})$ and then perform PCA via its eigen decomposition

$$\Sigma = V \Lambda V^\top, \tag{43}$$

where $\Lambda = \mathrm{diag}(\lambda_1, \dots, \lambda_d)$ contains the eigenvalues and $V = [v_1, \dots, v_d]$ contains the corresponding principal components. We denote the eigenvalues by $\{\lambda_k\}_{k=1}^d$ and the principal components by $\{v_k\}_{k=1}^d$.

During training, we periodically save model checkpoints. For each checkpoint at iteration $t$, we generate a batch of $B$ samples, flatten them into a matrix $\mathbf{X}_t \in \mathbb{R}^{B \times d}$, and compute the empirical covariance $\Sigma_t = \mathrm{Cov}(\mathbf{X}_t)$. Projecting onto the PCA basis $\{v_k\}_{k=1}^d$ of $\Sigma$ (see Eq.(43)), we measure the eigenvalues of the generated samples along mode $k$ at iteration $t$ as

$$\tilde{\lambda}_k(t) := v_k^\top \Sigma_t v_k, \qquad k = 1, \dots, d.$$

The trajectories $\{\tilde{\lambda}_k(t)\}_t$ provide a spectral view of how information along each principal direction is learned over time. It is worth note that the low-index mode (large eigenvalues $\lambda_k$) correspond to low-frequency, coarse features, whereas high-index mode correspond to high-frequency details (Wang & Pehlevan, 2026). Next, we quantify the emergence time of each eigenmode. Let $\tilde{\lambda}_k(t_1)$ denote the generated variance of mode $k$ at the first checkpoint $t_1$, and let $\lambda_k$ denote the corresponding eigenvalue, i.e., the $k$-th PCA eigenvalue of $\Sigma$. To quantify when the contribution of each mode becomes significant, we introduce a mode-dependent threshold, defined as the geometric mean between its initial and target eigenvalues,

$$\tilde{\lambda}_k^{\mathrm{th}} = \sqrt{\tilde{\lambda}_k(t_1)\, \lambda_k}.$$

The emergence time of eigenmode $k$ is then defined as the first training iteration at which its eigenvalue trajectory crosses this threshold in the direction of its target value:

$$t_k^* = \begin{cases} \min\{\, t : \tilde{\lambda}_k(t) \geq \tilde{\lambda}_k^{\mathrm{th}} \,\}, & \lambda_k > \tilde{\lambda}_k(t_1), \\ \min\{\, t : \tilde{\lambda}_k(t) \leq \tilde{\lambda}_k^{\mathrm{th}} \,\}, & \lambda_k < \tilde{\lambda}_k(t_1). \end{cases}$$

To visualize the global spectral learning trajectory of diffusion models, we collect all valid pairs $\{(t_k^*, k)\}$ and, for each emergence time $t$, summarize the set of indices $\{k : t_k^* = t\}$ by a single representative value. Plotting this value as a function of $t$ yields a smooth curve that captures how progressively higher-index modes are recruited as training proceeds (see Fig. 1b). We observe that low-index (large-eigenvalue, low-frequency) modes emerge early, while high-index (small-eigenvalue, high-frequency) modes only emerge much later in training, revealing a clear spectral bias in the learning dynamics of the diffusion model.

---

[1]Any $32 \times 32$ natural image dataset (e.g., CIFAR-10) can be used; the procedure is unchanged.

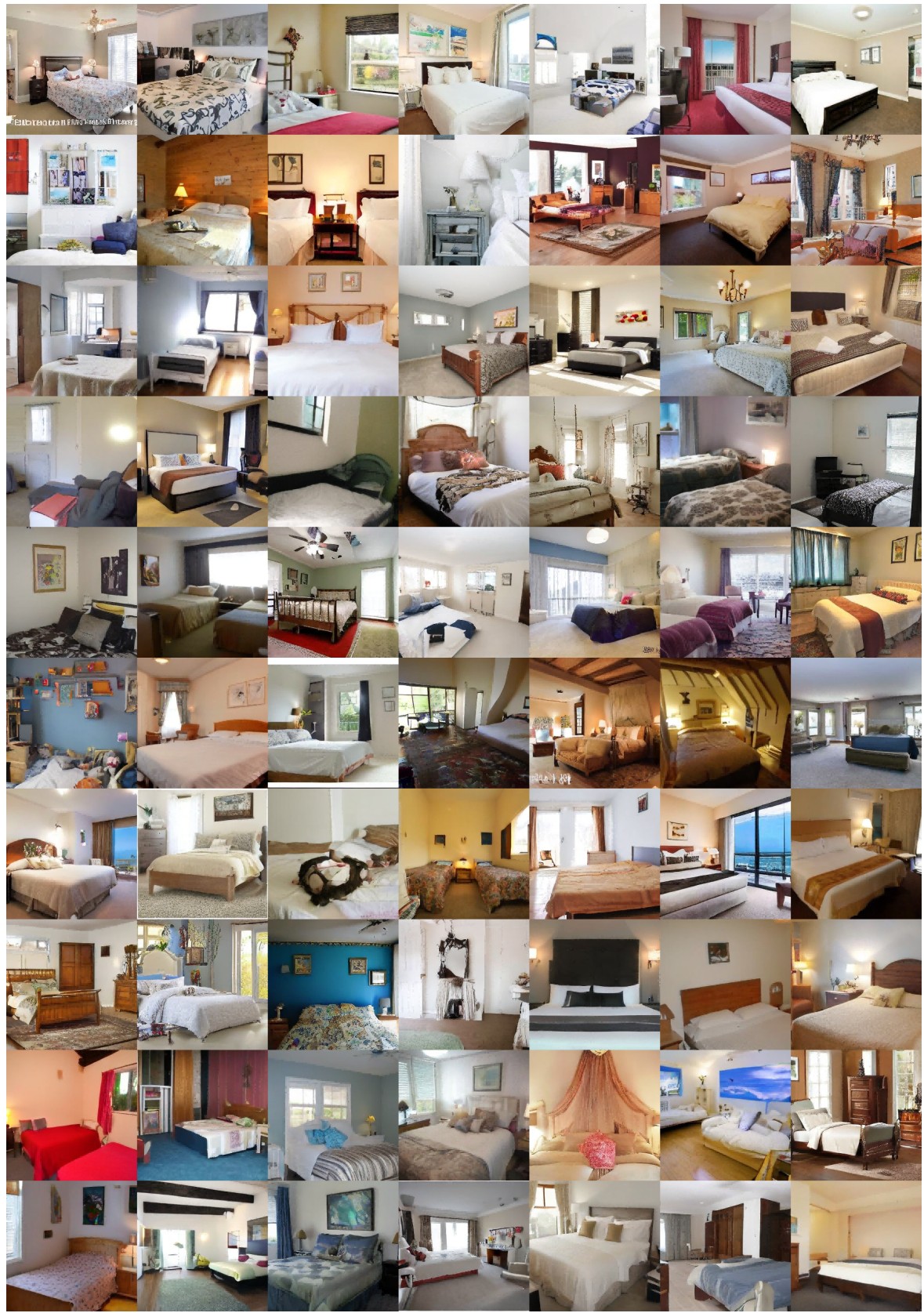

*Figure 8.* Randomly generated images from our method trained on LSUN-Bedroom-256×256. FID=2.72

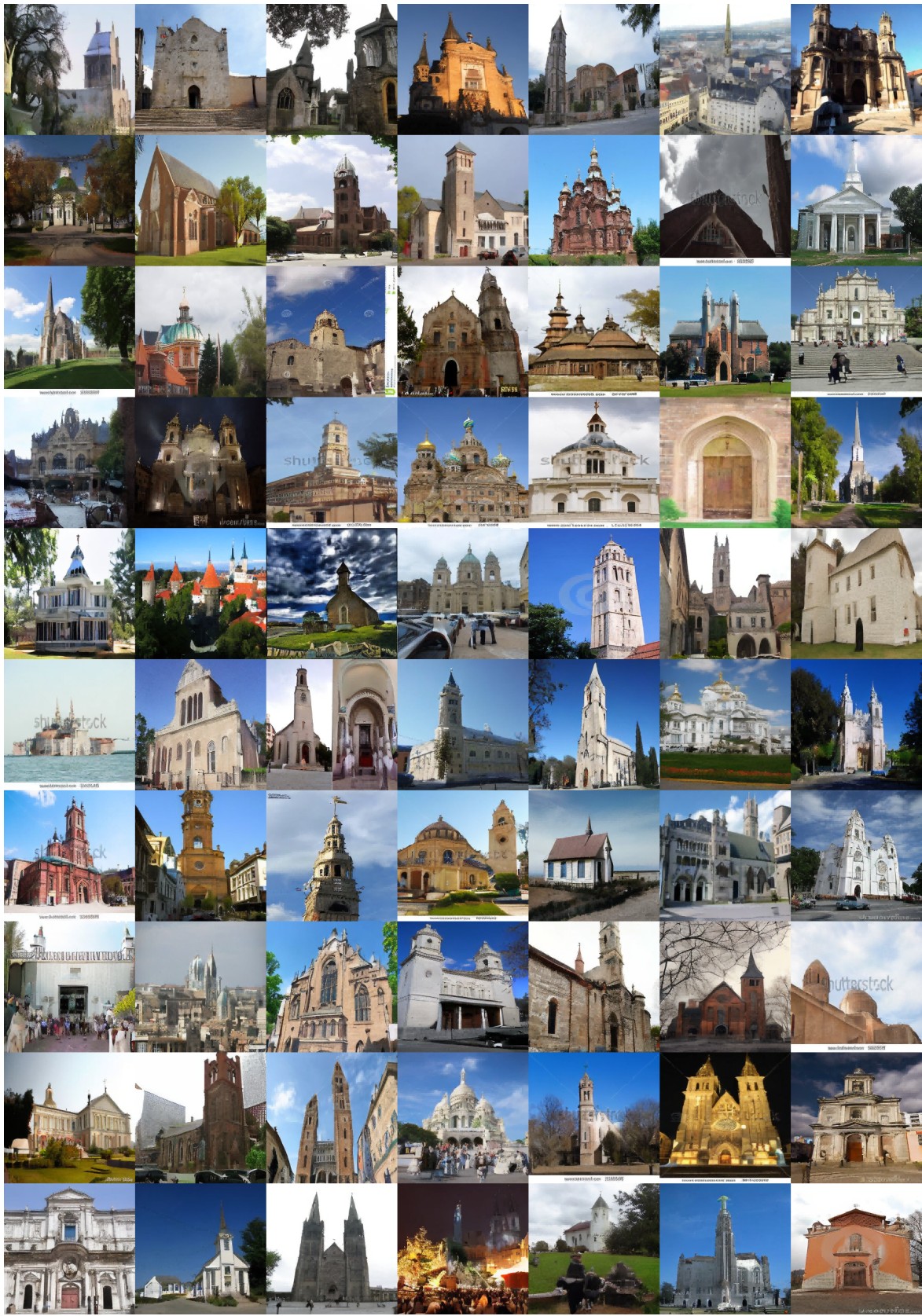

*Figure 9.* Randomly generated images from our method trained on LSUN-Church-256×256. FID=2.73

*Figure 10.* Randomly generated images from our method trained on FFHQ-64×64. FID=2.98

