# OpenReview forum: "FaPS: A General and Fast Training Method for Diffusion Models"
_ICML.cc/2026/Conference — ICML 2026 regular_

### Official Review · Reviewer_LueE · 2026-02-18

**Soundness:** 3
**Presentation:** 2
**Significance:** 3
**Originality:** 3
**Overall Recommendation:** 3
**Confidence:** 3

**Summary:**

This manuscript analyzes the existence of spectral bias in diffusion models along two dimensions. First, along the training iteration dimension, the model learns low-frequency components before high-frequency details. Second, along the diffusion timestep dimension, early denoising steps mainly reconstruct low-frequency content, while high-frequency details are progressively generated in later steps. Although I believe this phenomenon has been mentioned in other works (see [1]), after reading the paper carefully, I feel that this motivation was independently identified by the authors. Therefore, I still acknowledge that this paper provides a strong insight.

Based on this observation, the authors design an adaptive Frequency-Aware Gating Module, which dynamically selects the most informative patches for training according to the frequency content of each patch and the current diffusion timestep. Since patch selection is a discrete decision, they adopt policy gradient to address the non-differentiability issue. I find the method itself quite interesting. The writing is clear and easy to follow.

However, although I do like this method, the current experiments and experimental comparisons are completely disconnected from the recent development of diffusion training acceleration. There is a lack of comparison, similarity analysis, and experimental evaluation against state-of-the-art diffusion acceleration methods. I consider this a major weakness of the current experimental section. At ICML, this issue indeed prevents me from giving an accept recommendation. The detailed weaknesses and the additional experiments I believe should be included will be listed in the following weakness section. I hope the authors can address these concerns during the rebuttal period.

[1] *Improving the Diffusability of Autoencoders*, ICML 2025

**Compliance With Llm Reviewing Policy:**

Affirmed.

**Final Justification:**

I appreciate the authors’ response. In the latest reply, the authors clarified the comparison with Speed. I would also like to clarify my intent, as I mentioned in my rebuttal acknowledgment: the methods I listed were meant to provide you with more options, rather than requiring you to compare against all of them. This is also why I did not include these methods in my initial review. In fact, the acceleration performance of the newly listed methods is significantly weaker than LDM-based approaches.

Regarding the number of iterations, I noticed that the rebuttal does not specify the batch size, which makes it difficult to infer the number of epochs. The number of samples used for FID evaluation is also not stated. I think it would improve clarity and confidence if you explicitly mention something like: “Our evaluation strictly follows the settings in the original papers.”

I agree with the authors’ point that their method consumes less computation under the same number of iterations. However, strictly controlling training GFLOPs (as done in the RAE paper) would be more intuitive in my view. That said, based on the current experiments, FaPS combined with REPA-E achieves the best results, which is a very positive signal.

That being said, the model I originally had in mind to recommend was DiT-B/1. DiT-B/2, due to its smaller parameter size and patch size of 2, has been less commonly used in works after 2025. More importantly, I was hoping the authors could evaluate on DiT-XL/1. I understand this is difficult to complete within a short rebuttal period, but most prior works have reported results on this setting, which would allow readers like myself to more quickly assess the effectiveness of the method.

For the reasons above, my overall assessment of the current manuscript remains unchanged. However, given the authors’ response and additional experiments, I am willing to lower my confidence score and slightly increase other ratings. I also observe that the four reviewers have quite different opinions on this work. That means this work has clear strengths but also notable weaknesses. Nonetheless, I wish the authors the best of luck.

**Key Questions For Authors:**

I would like to ask why the methods selected for comparison in Table 1 are inference acceleration methods. Did I miss something?

**Limitations:**

The manuscript does not discuss limitations in much detail. The authors could, for example, comment on the potential latency introduced by the use of DFT.

They could also discuss whether the updates of θ and φ are stable in practice. Jointly updating two models often leads to imbalance issues, and it would be helpful to address whether this occurs in your setting.

**Strengths And Weaknesses:**

Strengths:

1. The theoretical motivation of the paper is strong. Although related ideas have appeared in previous work, the authors first establish a theoretical understanding of the learning dynamics of diffusion models through spectral analysis, and then design their method accordingly. The overall logic is coherent and self-consistent.

2. The writing is clear, and the figures are well presented. It is easy for readers to understand the authors’ intended message.

3. The method is orthogonal to existing approaches. The experiments show that FaPS can be combined with current acceleration methods such as SpeeD.

4. It is also interesting that FaPS improves generation quality under limited-data settings.

Weaknesses:

1. You mention in the Introduction that spectral bias along both dimensions has already been observed in prior work. I do believe this might also be your independent observation. However, you shouldn’t frame it as your core discovery. Instead, you should clearly explain how your empirical contribution goes beyond previous work, or clarify what exactly differentiates your analysis from existing findings.

2. I also find your statement *“These approaches focus on where to allocate computation, … but a key lever for further speedup is understanding what information diffusion models …”* hard to agree with. Patch Diffusion, at its core, selects training content through random patch sampling. Timestep reweighting methods (e.g., Min-SNR) also adjust which timesteps contribute more informative gradients. Aren’t these also decisions about “what information” to prioritize? This “what vs. where” framing feels like an oversimplification of prior work and makes the discussion seem less serious.

3. The Introduction should briefly explain how the *reward* is defined in the policy gradient formulation.

4. In Algorithm 1, the backpropagation steps are left as comments. More importantly, I could not find clear descriptions of the learning rates, optimizer choices, or update strategies anywhere in the paper. At minimum, these details should be included in the supplementary material.

5. Similarly, in Theorem 1, Assumption 1 requires the diffusion loss and the policy gradient terms to be bounded. However, when the noise follows a Gaussian distribution, the loss is theoretically unbounded. You should at least acknowledge and discuss this mismatch between theory and practice.

6. The comments above (1-5) are relatively **easy to fix** and mostly are clarification and presentation. However, the biggest concern in my view is the choice of baselines and comparisons. For example, in Table 1 you compare against DDPM, DDIM, NCSN++, and PNDM. As far as I understand, these are inference acceleration methods, while your method targets training acceleration. The goals are completely different. It’s unclear why they are being compared under the same setup.

7. Diffusion training acceleration is currently a very active research area. The manuscript includes almost no discussion or comparison with recent SOTA training acceleration methods (like VA-VAE [2], REPA [3], REPA-E [4], EQ-VAE [5]...). Given ICML’s high bar, I strongly suggest comparing against these methods. For example, try test fitting speed on ImageNet. It will make your claims more convincing. You mention having access to 8 V100 GPUs, so this should be feasible.

8. Finally, for the ablation studies, I suggest replacing the DFT-based frequency features with alternative patch features to test whether frequency awareness is truly necessary. One of your main claims is that frequency-aware patch selection uses computation more efficiently than random patch selection. This needs stronger empirical support.

[2] *Reconstruction vs. Generation: Taming Optimization Dilemma in Latent Diffusion Models*, CVPR 2025

[3] *Representation Alignment for Generation: Training Diffusion Transformers Is Easier Than You Think*, ICLR 2025

[4] *REPA-E: Unlocking VAE for End-to-End Tuning with Latent Diffusion Transformers*, ICCV 2025

[5] *EQ-VAE: Equivariance Regularized Latent Space for Improved Generative Image Modeling*, ICML 2025

---

> ### Author Rebuttal · Authors · 2026-03-31
>
> Thank you for your thoughtful comments and suggestions, and we try to address your questions and concerns below.
>
> **W1: You mention in the .. existing findings.**
>
> **A:** We agree that spectral bias has been noted in prior works and will modify our Introduction. While prior studies observed spectral bias across timesteps, our analysis formalizes a dual Spectral Bias (timesteps and training iterations). Crucially, we translate this mechanism into a learnable, frequency-aware gating policy to directly accelerate training.
>
> **W2: I also find your ... seem less serious.**
>
> **A:** We agree the 'what vs. where' framing oversimplifies prior work and will remove it in the revision. Our intended distinction is that existing methods like Patch Diffusion and Min-SNR treat data samples uniformly, optimizing spatial cropping or temporal weights without evaluating which specific image content is actually more important for the current training stage.
>
> **W3: The Introduction ... formulation.**
>
> **A:** We will add this clarification to the Introduction. Specifically, the equivalent 'reward' signal is the negative patch-level denoising loss ($-\ell_{patch}$).
>
> **W4: In Algorithm 1, the backpropagation ... supplementary material.**
>
> **A:** We clarify that Algorithm 1 includes the update steps for both the diffusion model (Line 14) and the gating policy (Line 18), alongside the comments. Additionally, the training details, including the optimizer (AdamW) and learning rate (1e-4), are provided in Appendix C.1. We agree that these details could be made more prominent. In the revision, we will include a comprehensive hyperparameter table for UNet and DiT.
>
> **W5: Similarly, in Theorem 1, ... between theory and practice.**
>
> **A:** We agree with this comment. Due to the Gaussian noise, the diffusion loss is unbounded. However, assuming bounded loss is a standard assumption in diffusion model (e.g., Lee et al. (2022)). Furthermore, since practical image data is normalized to $[-1, 1]$ and extreme noise values occur with exponentially vanishing probability, the empirical loss is well-behaved and does not exhibit unboundedness issues in practice. We will discuss this theory-practice gap in the revision.
>
> Lee et al (2022). Convergence for score-based generative modeling with polynomial complexity. NeurIPS 2022
>
> **W6 & Q1: The comments above (1-5) ... under the same setup.**
>
> **A:** We clarify that these models serve as reference baselines to demonstrate that our training acceleration does not compromise the final generative quality (FID) of standard U-Net frameworks. Our goal is to show that FaPS achieves better generative results with lower training cost; for a direct comparison with training-acceleration methods like Patch Diffusion (U-Net) and MDT (DiT), please refer to Tables 3-5.
>
> **W7: Diffusion training acceleration ... so this should be feasible.**
>
> **A:** We thank the reviewer and will include a discussion of these works. Notably, representation alignment methods (e.g., REPA/REPA-E) and latent space optimization (e.g., VA-VAE/EQ-VAE) are **orthogonal to our approach**. FaPS focuses on accelerating training from a **data perspective**, making it **naturally complementary** to these model-based methods. To validate this, we integrated FaPS with the method REPA-E. While fully training an ImageNet model to convergence is difficult during the rebuttal period, we conducted a fixed-budget experiment training all models for 100 GPU hours. As shown below, **REPA-E+FaPS** achieves a lower FID (47.16) and the highest speedup (~3.1x), suggesting FaPS can be combined with existing model-based methods for further acceleration. We will include the converged results in the revised version.
> | Method | FID | Speedup |
> | ----------- | ----- | ------- |
> | DiT-XL/2 | 85.53 | 1x |
> | EQ-VAE | 73.78 | ~1.3x  |
> | VA-VAE | 70.22 | ~1.3x |
> | REPA  | 55.63 | ~2.5x   |
> | REPA-E | 53.65 | ~2.7x |
> | REPA-E+FaPs | 47.16 | ~3.1x|
>
> **W8: Finally, for the ablation studies, ...  empirical support.**
>
> **A:** We appreciate the suggestion and clarify that this comparison is already in our main experiments (Table1-5). Specifically, our primary baselines, Patch Diffusion and MDT, essentially represent the 'random patch selection' strategies for U-Net and DiT architectures, respectively. The fact that FaPS outperforms both methods provides the exact empirical evidence requested: frequency-aware selection is better than random selection.
>
> **L: The manuscript ... whether this occurs in your setting.**
>
> **A:** We will add a Limitation part to discuss both points: (1) The computational overhead introduced by DFT is empirically negligible (see the response to W1 of Reviewer 8RdG). (2) We observe no instability during joint training. Besides the module ($\phi$) being lightweight, this stability is mathematically backed by Theorem 1: the joint optimization objective is well-bounded, ensuring gradient updates for both $\phi$ and the main network $\theta$ remain aligned.

---

> > ### Author Rebuttal · Reviewer_LueE · 2026-04-01
> >
> > I sincerely appreciate the authors’ response. For W1–W6, I am generally satisfied with the answers, and my concerns have been addressed. The authors have also committed to making revisions, which I believe will help improve the quality of the manuscript.
> >
> > However, regarding the response to W7, I find it difficult to agree with the authors’ current position. The reasons are as follows:
> >
> > **1.** As I mentioned, FaPS appears completely disconnected from recent progress in accelerating diffusion training. From Tables 2 and 4, we can see that the compared methods are almost all from 2024 or earlier. However, this area has become a very active topic recently, with many influential works emerging in 2025 and 2026. For example, the ICML 2025 papers I previously recommended, EQ-VAE [1] and SE [2].
> >
> > **2.** Beyond these two methods, there are also several works from other top venues: VA-VAE [3], REPA [4], RAE-DITDH [5], REPA-E [6], SVG [7], and DDT [8] have not been compared against. I acknowledge the effort made in W7. However, the issue is that the metrics for these methods are readily available (e.g., Table 8 in RAE). Since your experiments are not conducted under the same backbone and dataset settings, FaPS remains disconnected from the current progress in training acceleration.
> >
> > **3.** For the same reason, I am also skeptical about the results reported in W7. Specifically:
> >
> > 1. The 100 GPU-hour constraint is too weak; such problems are typically controlled by the number of training epochs.
> > 2. The reported FID values are unexpectedly high. For example, VA-VAE achieves 2.11 at 64 epochs, yet your table reports 70.22. Moreover, the speedup of VA-VAE over DiT-XL/2 cannot plausibly be only 1.3×. I believe this comes from inconsistent training and evaluation settings. Therefore, in my view, the table in W7 has very low credibility. If included in the final paper, this could raise serious academic integrity concerns.
> >
> > **4.** I suggest the authors strictly control experimental variables. If computational resources are limited, you could use DiT-B (~200M parameters, which trains relatively fast with FP16), though DiT-XL is preferable since most methods report results on that backbone. You should train on the same ImageNet dataset with the same number of epochs and report FID-50K (since FID varies significantly with the number of generated samples). Given precedents like EQ-VAE and SE, I consider this a baseline requirement for acceptance at ICML.
> >
> > **5.** If you are uncomfortable with LDM-based methods, I also recommend comparing with recent works such as JiT [9], DeCo [10], PixelDiT [11], PixelGen [12], and EPG [13]. I understand this is a long list, but I am not asking you to compare with all of them. You can prioritize those most relevant. However, at minimum, there should be at least one experiment with strictly aligned settings.
> >
> > **6.** Regarding your claim of orthogonality: for example, REPA is orthogonal to SVG and RAE, yet those works still provide direct comparisons with REPA. This shows that orthogonality is not a valid reason to avoid direct comparison. Of course, if you can demonstrate that FaPS, with minor modifications, can further improve upon these methods (which I believe is possible), that would significantly strengthen your claims.
> >
> > Based on the above concerns, without direct comparisons, the strongest recommendation I can give at this stage is a weak reject. I genuinely hope the authors take the time to consider these suggestions rather than rushing. Doing so would make your work much stronger!
> >
> > [1] *EQ-VAE: Equivariance Regularized Latent Space for Improved Generative Image Modeling, ICML 2025*
> >
> > [2] *Improving the Diffusability of Autoencoders, ICML 2025*
> >
> > [3] *Reconstruction vs. Generation: Taming Optimization Dilemma in Latent Diffusion Models, CVPR 2025*
> >
> > [4] *Representation Alignment for Generation: Training Diffusion Transformers Is Easier Than You Think, ICLR 2025*
> >
> > [5] *Diffusion Transformers with Representation Autoencoders, ICLR 2026*
> >
> > [6] *REPA-E: Unlocking VAE for End-to-End Tuning with Latent Diffusion Transformers, ICCV 2025*
> >
> > [7] *Latent Diffusion Model Without Variational Autoencoder, ICLR 2026*
> >
> > [8] *DDT: Decoupled Diffusion Transformer, CVPR 2026*
> >
> > [9] *Back to Basics: Let Denoising Generative Models Denoise*
> >
> > [10] *DeCo: Frequency-Decoupled Pixel Diffusion for End-to-End Image Generation, CVPR 2026*
> >
> > [11] *PixelDiT: Pixel Diffusion Transformers for Image Generation*
> >
> > [12] *PixelGen: Pixel Diffusion Beats Latent Diffusion with Perceptual Loss*
> >
> > [13] *EPG: There is No VAE: End-to-End Pixel-Space Generative Modeling via Self-Supervised Pre-Training, ICLR 2026*

---

> > > ### Author Response · Authors · 2026-04-07
> > >
> > > **A1 & A2:**  We thank the reviewer for emphasizing the importance of relating FaPS to recent diffusion acceleration works. We would like to clarify: **(1)** Our evaluation already includes recent 2025 baselines such as Speed; the comparisons are not limited to methods from 2024 or earlier. Please refer to Table 4 in our paper.  **(2)** While several of the methods mentioned by the reviewer were mainly evaluated on specific architectures (e.g., DiT), FaPS is designed to be architecture-agnostic, and we have demonstrated its effectiveness on both U-Net and DiT backbones (Table 1-3 for U-Net and Table 4-5 for DiT). **(3)** The 2026 works referenced by the reviewer are just formally published/accepted very close to or after the ICML submission deadline (which are usually considered as concurrent work under standard conference guidelines). Nevertheless, we still want to try our best to address the reviewer's concern, even under  the limited rebuttal period. We conducted a partial comparison with a recent method REPA-E[1], as summarized in A5/A6 below. We hope these newly conducted experimental results could be helpful.
> > >
> > > [1] REPA-E: Unlocking VAE for End-to-End Tuning with Latent Diffusion Transformers, ICCV 2025
> > >
> > > **A3 & A4:** We thank the reviewer for their comments and would like to clarify some misunderstandings regarding the experiment settings and results.
> > >
> > > **(1):** Controlling by 'training epochs' is not quite suitable for evaluating patch-selection acceleration methods like FaPS, Patch Diffusion (Wang et al.,2023), and MDT (Gao et al., 2023).  Since these methods select only **a subset of patches** from each image during training, **their computational cost per epoch is lower than that of standard methods  (e.g., DDPM, DiT).**  As a result, under the same wall-clock time budget, they can complete more epochs than conventional training approaches. We follow **a fixed wall-clock time** budget (the same as Wang et al.,2023; Gao et al., 2023), since it can provide a more direct and fair measure of training efficiency.
> > >
> > > **(2):**  The reviewer mentioned the comparison between a converged FID of 2.11 (achieved at 64 epochs, approximately **3,300K iterations**) and our 100 GPU-hour early-stage snapshot (approximately **55K iterations**). However, the results obtained at such vastly different training scales are **not directly comparable**. At this ~55K iteration setting, our results are in close with the original works; for example, the reported FID for EQ-VAE is 73.78, which is similar to the FID of 73.6 reported at 50K iterations in the original EQ-VAE paper[3]. This indicates that our experimental setup and results are consistent with the original works at comparable early-stage iterations. Also, the observed ~1.3× speedup reflects the actual wall-clock improvement under the same hardware setup.
> > >
> > > Finally, as noted in our previous response, the 100-hour table was intended to demonstrate early-stage acceleration trends during the **limited rebuttal period**. We will additionally provide **converged** experimental results in the revised manuscript.
> > >
> > > [1] Wang et al. "Patch diffusion: Faster and more data-efficient training of diffusion models." NeurIPS 2023.
> > >
> > > [2] Gao et al. "Masked diffusion transformer is a strong image synthesizer."  ICCV 2023.
> > >
> > > [3] Kouzelis et al. "EQ-VAE: Equivariance Regularized Latent Space for Improved Generative Image Modeling". ICML 2025
> > >
> > > **A5 & A6:** Following the reviewer's suggestion, we conducted experiments on the **DiT-B/2** (130M) backbone. The results, summarized below, show that FaPS reduces training time while maintaining competitive FID. Moreover, combining FaPS with REPA-E further improves both FID and speedup, suggesting that FaPS is effective on its own and can complement existing acceleration methods.
> > >
> > > | Method               | Iterations | Training Times (h) with 8 GPUs | FID       | Speedup |
> > > | -------------------- | ---------- | ------------------------------ | --------- | ------- |
> > > | DiT-B/2              | 400K       | ~43h                           | 34.52     | 1x      |
> > > | REPA-E               | 400K       | ~43h                           | 23.21     | ~2.1x   |
> > > | FaPS (not converged) | 400K       | ~25h                           | 35.63     | -       |
> > > | **FaPS**             | 690K       | ~43h                           | 24.59     | ~1.9x   |
> > > | **REPA-E + FaPS**    | 690K       | ~43h                           | **21.07** | ~2.5x   |

---

### Official Review · Reviewer_8RdG · 2026-02-25

**Soundness:** 3
**Presentation:** 3
**Significance:** 3
**Originality:** 3
**Overall Recommendation:** 5
**Confidence:** 2

**Summary:**

This paper proposes a fast training method for diffusion models, named Frequency-aware Patch Selection (FaPS), whose core is based on the authors' observation of the "dual spectral bias" in the learning dynamics of diffusion models. Empirical analysis of diffusion training reveals that (1) low-frequency components are fit earlier than high-frequency details, (2) early denoising steps mainly focus on coarse low-frequency content, while later steps complete high-frequency details. To leverage these, FaPS introduces a lightweight frequency-aware gating module. This module adaptively selects the most informative image patches for computation based on their frequency characteristics and the current time step, thereby avoiding computational waste on redundant or already learned content. Since block selection involves discrete, non-differentiable decisions, the authors model it as a stochastic policy network. They employ policy gradient methods for end-to-end joint optimization and theoretically prove the convergence of this process. Experimental results demonstrate that FaPS achieves over 3x training acceleration while maintaining or improving generation quality. It also exhibits exceptional data efficiency under limited data, lowering the barrier for researchers to train diffusion models.

**Compliance With Llm Reviewing Policy:**

Affirmed.

**Final Justification:**

The authors’ rebuttal successfully addressed my primary concerns regarding computational overhead and the empirical validation of the "dual spectral bias." I maintain my positive recommendation for this submission.

**Key Questions For Authors:**

1. Can the authors provide quantitative or qualitative analyses of the gating module’s patch selection dynamics in actual training? For example, are patches with higher frequency content progressively more likely to be selected at later training stages or smaller timesteps, as hypothesized?
2. How consistent are speedup and FID results across multiple random seeds or training runs? Can the authors report variance or confidence for key summary metrics in Tables 1–5?
3. Can the authors clarify whether batch size, epochs, and hardware were kept identical across all baselines when calculating speedup ratios? Otherwise, how were timing figures normalized?
4. In high-resolution settings (256x256 in paper or even higher in SOTA models), does selecting only a single patch per iteration lead to any artifacts or lack of global coherence? Have you observed any need for a curriculum that increases the number of selected patches or the patch size as training progresses?

**Limitations:**

Yes

**Strengths And Weaknesses:**

Strengths
Soundness:
The paper moves beyond heuristic-based patch selection in diffusion model training, proposes a trainable router that explicitly leverages frequency data and timestep to decide which patches to prioritize, making the process more adaptive than traditional methods. The paper also provides a non-convex convergence guarantee to prove the joint optimization with a stochastic policy network remains theoretically well-behaved. The method is validated across multiple datasets, model architectures, and compared against a comprehensive collection of SOTA methods.
Presentation:
The paper follows a clear progression from empirical observations of “dual spectral bias” to a well-structured and detailed solution.
Significant:
The method achieves up to 3x training acceleration while maintaining performance, addressing a key pain point in diffusion model training. The paper also presents performance improvement in a limited-data setting, which is of high practical value in real-world conditions.
Originality
The identification of “dual spectral bias” in both training iterations and diffusion timesteps is a novel insight. Modeling patch selection as a stochastic policy trained via policy gradients is an innovative combination in the domain.
Weaknesses
Soundness:
Even though the overall acceleration is counted, the overhead of the gating module and DFT operations is not benchmarked. Also, there is little analysis or visualization of the actual patch selection over training iterations. The speedup and FID metrics are all reported as single values, and the conclusion would be less convincing if the numbers are based on single runs. The selection of hyperparameters (learning rate, number of patches, etc.) is not fully explored.
Presentation:
The paper leaves parts of critical literature review and PCA mode analysis in the appendix, making the claims less grounded.
Significance:
The method is validated at a maximum resolution of 256x256, which represents a significant discrepancy compared to SOTA models typically operating at 512×512, 1024×1024, or even higher resolutions. Further investigation is needed to determine whether the methodology maintains structural integrity or introduces artifacts when scaled to higher-resolution.
Originality:
While the combination is novel within the domain, the individual components proposed in this paper are already well-established, making the work more like an incremental application of RL to diffusion models.

---

> ### Author Rebuttal · Authors · 2026-03-31
>
> Thank you for your thoughtful comments and suggestions, and we try to address your questions and concerns below.
>
> **W1 & Q2: Soundness.**
>
> **A:**
> **(1)** We agree that providing explicit benchmarks for the overhead offers a more precise assessment. While our reported $3\times$ speedup is measured in total wall-clock time, we provide a detailed per-iteration time breakdown in the table below to address your concern.
>
> **Table: Time breakdown per training iteration**
> | Computational Component | Time per Iteration (ms) | Proportion|
> | -------------------------------- | ----------------------- | --------------- |
> | Standard U-Net | $\sim$9760 ms | 100% (Baseline) |
> | **FaPS:** DFT Feature Extraction | $\sim$10 ms | $\sim$0.1%  |
> | **FaPS:** Gating Module | $\sim$110 ms | $\sim$1.1% |
>
> As shown, the entire gating mechanism introduces an overhead of **<1.2%**, which is negligible. We will add this table to the revision.
>
> **(2)** As noted in our response to Question 1, we have provided an analysis of the actual patch selection over the training iterations.
>
> **(3)** Due to the massive computational cost of training diffusion models from scratch (e.g., ~13 days on 32 A100 GPUs for a single baseline run), conducting multiple runs with different seeds is computationally prohibitive. We notice that in the previous DDPM and DiT papers, they usually also did not perform multiple runs. To  provide empirical evidence for the reliability of our method, we conducted the experiments over different datasets and achieve consistent speedup and FID improvements across **six diverse datasets**.
>
> **(4)** To ensure a fair comparison, we did not tune the hyperparameters. Since our approach is built upon existing baselines, we directly adopted the optimal hyperparameter settings (e.g., learning rate, number of patches) established by the original papers.
>
> **W2: Presentation**
>
> **A:** We thank the reviewer for the structural feedback. In the revision, we will integrate the important literature review and the PCA mode analysis into the main text to strengthen our claims.
>
> **W3: Significance**
>
> **A:** We agree that evaluating at higher resolutions (e.g., 512x512) is necessary. Due to the short rebuttal period and the high computational cost of diffusion training, we unfortunately cannot complete these experiments right now. Theoretically, since higher-resolution images contain greater spatial redundancy, our patch-selection method is more likely to yield even higher speedups at these scales. We will include the high-resolution results in the revised manuscript.
>
> **W4: Originality**
>
> **A:** We clarify that our contribution lies in the discovery of the **dual Spectral Bias** phenomenon during diffusion training, and in the design of an efficient adaptive policy (**FaPS**) that leverages this insight to accelerate training. Reinforcement learning is just the tool to implement this frequency-aware patch selection strategy, which is not the major direction that we intend to improve.
>
> **Q1: Can the authors...  as hypothesized?**
>
> **A:** To quantify the selection dynamics, we sampled a random batch and divided all patches equally (1:1) into High-Frequency (HF) and Low-Frequency (LF) groups based on their frequency information. We then calculated the proportion of HF patches selected by our gating module, with random selection serving as a baseline (50%). As shown in the table below, the module favors LF patches at smaller timesteps and increasingly selects HF patches at later timesteps and later training stages, which provides empirical support for our dual Spectral Bias hypothesis. This table will be included in the revision.
>
> **Table: Proportion of High-Frequency Patches in the Selected Subset (%)**
> | Timestep | Early Training Stage (1/3) | Late Training Stage (2/3) |
> | -------- | -------------------------- | ------------------------- |
> | 300| 42% | 55% |
> | 700| 52% | 67%|
>
> **Q3: Can the authors ... timing figures normalized?**
>
> **A:** Yes. All hardware configurations (e.g., A100 GPUs) and batch sizes were kept strictly identical across our method and all baselines to ensure a fair comparison. The $3\times$ speedup is the absolute reduction in total wall-clock time required to reach the same target FID; therefore, no timing normalization was needed.
>
> **Q4: In high-resolution settings ... as training progresses?**
>
> **A:** To ensure global coherence, our method adopts the mechanism of Patch Diffusion, where the selected patch size progressively increases during training. We will add a detailed explanation of this dynamic sizing mechanism in the revised manuscript.

---

> > ### Author Rebuttal · Reviewer_8RdG · 2026-04-01
> >
> > I appreciate the authors for their detailed and professional rebuttal. Most of my concerns are sufficiently addressed. However, I have a remaining concern regarding the Proportion of High-Frequency Patches table, which appears to contradict the paper’s core theoretical framework:
> >
> > - The manuscript argues that early denoising steps (large t) focus on low-frequency structures. However, the table shows that at t=700, the module selects a higher proportion of high-frequency (HF) patches (52%–67%) than at t=300 (42%–55%). The rebuttal text states the module "favors LF patches at smaller timesteps". This is diametrically opposed to the "Dual Spectral Bias" hypothesis, which posits that low-frequency components are fit earlier in training and at larger timesteps.
> >
> >
> > Could the authors clarify if this is a labeling error or if the learned gating policy is actually operating in a manner that contradicts the proposed frequency-based theory?

---

> > > ### Author Response · Authors · 2026-04-02
> > >
> > > We thank the reviewer for pointing this out.  This is indeed a labeling error in our previous response, which does not contradict the paper’s core theoretical framework. We have provided the corrected table below:
> > >
> > > | **Timestep**                   | **Early Training Stage (1/3)** | **Late Training Stage (2/3)** |
> > > | ------------------------------ | ------------------------------ | ----------------------------- |
> > > | **700 (Early Denoising Step)** | 42%                            | 55%                           |
> > > | **300 (Late Denoising Step)**  | 52%                            | 67%                           |
> > >
> > > This behavior is consistent with the Dual Spectral Bias hypothesis. We apologize for the confusion and will include this corrected Table in the revision.

---

### Official Review · Reviewer_gnjU · 2026-03-14

**Soundness:** 3
**Presentation:** 3
**Significance:** 3
**Originality:** 3
**Overall Recommendation:** 4
**Confidence:** 5

**Summary:**

The paper proposes FaPS (Frequency-aware Patch Selection), a training method designed to accelerate diffusion model training by leveraging the dual spectral bias observed in diffusion models. The authors empirically show that diffusion models tend to learn low-frequency components earlier than high-frequency details, both across training iterations and along diffusion timesteps. Motivated by this observation, FaPS introduces a frequency-aware gating module that adaptively selects informative image patches based on their frequency features and the current diffusion timestep, allowing the model to focus computation on the most useful regions. Since patch selection is a discrete decision, the gating module is formulated as a stochastic policy network and optimized using policy gradient methods to enable end-to-end training. Experiments on several datasets and architectures (e.g., UNet and DiT) show that FaPS can achieve up to $3 \times$ faster training while maintaining comparable or improved generation quality.

**Compliance With Llm Reviewing Policy:**

Affirmed.

**Final Justification:**

Overall, I remain positive about this paper. The paper is clearly written and presents an interesting and potentially useful training strategy for accelerating diffusion models. My main concerns were the practical informativeness of the convergence guarantee and the relatively limited diversity and recency of the empirical baselines; the rebuttal addressed these concerns partially by clarifying the intent and scope of the theory and by providing additional evidence and discussion, but it did not substantially change my overall evaluation.

**Key Questions For Authors:**

**Q1.** Compared to the standard denoising training objective for diffusion models, why does the training loss in Eq. (4) not involve integration (or expectation) over the diffusion timestep?

**Q2.** The paper claims up to 3× faster training, but it is unclear how this improvement is quantified. Is the comparison based on wall-clock time, GPU hours, or the number of training iterations under identical hardware settings?

**Limitations:**

Yes

**Strengths And Weaknesses:**

**Strengths**:

$\bullet$ 1. The paper is generally well written and clearly organized. The motivation, method description, and experimental setup are presented in a clear and easy-to-follow manner.

$\bullet$ 2. The work combines both theoretical analysis and empirical evaluation. In particular, the authors provide a convergence analysis for the proposed method and validate its effectiveness through experiments across multiple datasets and model architectures.

**Weaknesses**:

$\bullet$ 1. The convergence guarantee appears to be relatively weak. The result only shows that the average gradient norm is bounded by a constant determined by the gradient noise variance, implying convergence to a neighborhood of a stationary point rather than ensuring $\|\mathbb{E}[\nabla L(w)]] \to 0$. In particular, since the gradients of the gating module are estimated via policy gradient, which can have high variance, it is unclear whether the bounded-variance assumption required by the theorem is realistic in practice.

$\bullet$ 2. The experiments are conducted on several commonly used generative modeling benchmarks, including CelebA, FFHQ, LSUN, MetFaces, AFHQv2, and MS-COCO. While these datasets cover multiple settings (low-resolution, high-resolution, and limited-data), many of them are face-centric or relatively structured, which may not fully reflect the diversity of real-world image distributions. Additional evaluation on more diverse large-scale datasets, such as ImageNet, could further strengthen the empirical validation of the proposed method.

$\bullet$ 3. The choice of baselines is generally reasonable, as the experiments include widely used diffusion training frameworks such as EDM with DDPM++/ADM and also evaluate the method on both UNet and DiT backbones. However, the comparisons are primarily against diffusion training pipelines proposed in 2022 or 2023. Including stronger efficiency-oriented baselines in 2024, 2025, or 2026 would better contextualize the claimed training speed improvements. Here are some potential choices:

[1] Yu, S., Kwak, S., Jang, H., Jeong, J., Huang, J., Shin, J., & Xie, S. Representation Alignment for Generation: Training Diffusion Transformers Is Easier Than You Think. In ICLR 2026.

[2] Yao, J., Yang, B., & Wang, X. Reconstruction vs. generation: Taming optimization dilemma in latent diffusion models. In CVPR 2025.

$\bullet$ 4. Some minor issues:

 - Line 181: SDE should be PDE

I am willing to raise the rating if the authors could address the above weaknesses.

---

> ### Author Rebuttal · Authors · 2026-03-31
>
> Thank you for your thoughtful comments and suggestions, and we try to address your questions and concerns below.
>
> **W1: The convergence guarantee appears to be relatively weak.**
>
> **A:** We thank the reviewer for the thorough theoretical review.
>
> **(1):** Under a fixed step size ($\max \\{ \eta_\theta, \eta_\phi \\} \le 1/L$), this result naturally follows from the algorithm: the average gradient norm converges to a neighborhood determined by the gradient noise. This is the standard convergence result for stochastic gradient-based optimization in non-convex settings (e.g., Bottou et al., 2018). Achieving exact asymptotic convergence to zero ($\mathbb{E}[\\|\nabla L(w)\\|] \to 0$) would require adopting a diminishing learning rate schedule. We chose to present the constant step-size bound because it better reflects the practical training regimes of diffusion models. We will add a brief remark in the theoretical section to clarify that convergence can be obtained under a diminishing schedule.
>
> **(2):** While vanilla REINFORCE suffers from high variance, we utilize an Exponential Moving Average (EMA) baseline $b$ to reduce it. Theoretically, Assumption 1 (Appendix B) bounds the patch loss $\ell_k(\theta; z)$ by a constant $B_\ell$. Since the policy gradient is weighted by this bounded loss offset by a bounded baseline, the estimator's variance ($\sigma_\phi^2$) is bounded. Thus, our bounded-variance assumption is reasonable. We will connect Assumption 1 and the EMA baseline in Appendix B.
>
> Bottou et al. (2018). Optimization methods for large-scale machine learning. *SIAM review*.
>
> **W2: The experiments... the proposed method.**
>
> **A:** We agree that evaluating on diverse datasets like ImageNet is important. Fully training an ImageNet model is difficult during the rebuttal period, so we conducted a fixed-budget experiment (100 GPU hours). As shown in the table below (see W3), FaPS converges faster and achieves a lower FID compared to the baselines, which suggests its effectiveness on highly diverse real-world distributions. Fully converged results will be included in the revision.
>
> **W3: The choice of baselines is generally reasonable,.. are some potential choices.**
>
> **A:** We thank the reviewer for highlighting these recent baselines. Crucially, methods like Representation Alignment [1] and Latent Space Optimization [2] focus on architectural or feature-level guidance. In contrast, FaPS accelerates training from data perspective. Therefore, our method is **orthogonal and complementary** to these methods. To empirically validate this, we integrated FaPS with the recent method REPA-E under the aforementioned 100 GPU-hour ImageNet setting. As shown in the table below, **REPA-E+FaPS** demonstrates the lowest FID (47.16) and the highest speedup (~3.1x), compared with REPA-E alone (FID 53.65, ~2.7x speedup). This suggests that FaPS can be effectively combined with recent baselines for further acceleration. A comprehensive discussion and these comparisons will be included in the revised manuscript.
>
> | **Method**             | **FID**   | **Speedup** |
> | ---------------------- | --------- | ----------- |
> | DiT-XL/2               | 85.53     | 1x          |
> | EQ-VAE                 | 73.78     | ~1.3x       |
> | VA-VAE [2]             | 70.22     | ~1.3x       |
> | REPA [1]               | 55.63     | ~2.5x       |
> | REPA-E                 | 53.65     | ~2.7x       |
> | **REPA-E+FaPs (Ours)** | **47.16** | **~3.1x**   |
>
> **W4: Some minor issues.**
>
> **A:** We thank the reviewer for the careful reading. We suspect there might be a minor typo in the comment, and the reviewer likely meant **"ODE"** instead of "PDE". If so, we completely agree with the reviewer's intuition. Equation (3) is indeed an ODE (the Probability Flow ODE), as it does not contain a stochastic Wiener process term ($dw$). We will revise the sentence structure in the updated manuscript to clearly separate the two concepts (ODE and SDE).
>
> **Q1: Compared to ... the diffusion timestep?**
>
>  **A:** We appreciate the reviewer’s careful attention to this point. Equation (4) does not include the expectation over the timestep $t$ because it is formulated to represent the expected $L_2$ denoising error **independently for a specific, given noise level $\sigma_t$**. We adopted this specific notation to stay consistent with the continuous-time framework of EDM (Karras et al., 2022), where the denoiser is strictly defined to minimize the score matching error at each noise scale independently.
>
> Karras et al., (2022). Elucidating the design space of diffusion-based generative models. NeurIPS 2022.
>
> **Q2: The paper claims ... under identical hardware settings?**
>
> **A:** The $3\times$ speedup is quantified as the reduction in wall-clock training time (GPU hours) required to achieve the same FID under the same experimental setting (see line 348 of our manuscript).

---

> > ### Author Rebuttal · Reviewer_gnjU · 2026-04-03
> >
> > Thanks for the author's reply. For the answer to W1, the rebuttal clarifies why exact convergence is not claimed under constant step sizes, but it does not resolve the main concern that the guarantee may be too weak to be practically informative. Nevertheless, I remain positive about this paper.

---

> > > ### Author Response · Authors · 2026-04-03
> > >
> > > We thank the reviewer for the positive feedback and agree with the comment. As is common in theoretical analyses of deep generative models, the current guarantee is relatively loose and is intended primarily to provide theoretical support rather than a tight characterization of practical training behavior. We appreciate this important point and will include a discussion in the limitations section of the revised manuscript to clarify the gap between the theoretical bound and its practical informativeness.

---

### Official Review · Reviewer_okjv · 2026-03-14

**Soundness:** 2
**Presentation:** 1
**Significance:** 3
**Originality:** 2
**Overall Recommendation:** 4
**Confidence:** 3

**Summary:**

This paper proposes FaPS (Frequency-aware Patch Selection), a training strategy for patch-wise diffusion models that adaptively selects the most informative patch at each training step based on its frequency characteristics and the current diffusion timestep, instead of uniformly sampling patches. To do this, the authors introduce a lightweight frequency-aware gating module that predicts patch selection probabilities and train the discrete patch-selection policy end-to-end with a REINFORCE-style objective, alongside a theoretical convergence analysis. Across UNet-based patch diffusion, transformer-based settings, and text-to-image generation, the paper shows that FaPS can substantially accelerate training while maintaining or improving sample quality, and is especially helpful in data-limited regimes.

**Compliance With Llm Reviewing Policy:**

Affirmed.

**Key Questions For Authors:**

-

**Limitations:**

-

**Strengths And Weaknesses:**

The paper proposes an interesting and potentially impactful idea: adaptive patch selection for patch-wise diffusion training. I found this core contribution novel and practically meaningful, especially because it targets a real inefficiency in patch-based training—namely, that not all patches are equally useful at all stages of optimization. In that sense, the paper goes beyond a purely engineering tweak: it suggests that patch selection itself can be learned, and that doing so can improve training efficiency while preserving or even improving generation quality. Even though I have substantial reservations about how the motivation is developed, I still think the underlying idea is strong enough to make the work worth accepting, particularly if the authors are willing to substantially revise the presentation and sharpen the conceptual framing.

My main concern is that the motivation and empirical narrative do not currently form a fully convincing logical chain. In particular, Figure 1-(a) and Figure 1-(b) appear to juxtapose two different phenomena: frequency evolution across sampling timesteps and training iterations—without clearly establishing why these should jointly imply the proposed adaptive patch-selection strategy. These are different axes of analysis, and the paper does not adequately justify the transition from these observations to the specific algorithmic design. What's more, the use of PCA eigenmode index seems at best an indirect proxy for frequency behavior; it looks closer to a notion of mode coverage or variance capture than a direct measure of spectral learning. As a result, while I can understand the authors’ intuition that coarse-to-fine behavior may implicitly motivate frequency-aware patch selection, I do not think the current exposition makes that argument rigorously enough. It's about the flow of the paper writing, not about the algorithm itself.

A second conceptual issue is that the actual learning objective seems more naturally interpreted as adaptive loss-driven patch selection than as principled frequency-aware selection. The method may still work well in practice, but the paper’s current framing overstates the extent to which the learned policy is truly tied to frequency structure, rather than simply discovering which patches are currently more favorable for optimization. Along similar lines, I also found the policy-learning direction somewhat counterintuitive: under the stated objective, the method appears to prefer lower-loss patches rather than explicitly emphasizing harder or more informative ones, which raises questions about whether the intended mechanism is best understood as curriculum learning, easy-example preference, or something else. Despite these weaknesses, I would still lean weak accept, because the central idea is interesting, experimentally promising, and potentially valuable to the community. My recommendation is therefore acceptance conditional on a major revision, where the authors substantially improve the motivation, clarify the role of the eigenmode/frequency analysis, and more honestly align the conceptual claims with what the optimization objective is actually doing.

---

> ### Author Rebuttal · Authors · 2026-03-31
>
> Thank you for your thoughtful comments and suggestions, and we try to address your questions and concerns below.
>
> **W1: My main concern ... It's about the flow of the paper writing, not about the algorithm itself.**
>
> **A:** We thank the reviewer for this helpful comment on the flow of the paper.
>
> **(1):**  We agree that the transition from observing the "dual spectral bias" to the specific design of **FaPS** needs stronger explicit justification in the text. In the revision, we will clarify this logical chain: The fact that frequency preference evolves across *two distinct axes* (both the diffusion **timestep** and the **training iteration**) is the reason why we designed FaPS as an **adaptive** policy network. If the spectral bias evolved along only one axis, a static, hand-crafted strategy might suffice (please see challenge **C1** in line 125). However, the usefulness of a patch varies throughout training and depends on both the current training iteration and the diffusion timestep, making static heuristics suboptimal. This requires a trainable gating module that **adaptively** selects patches based on their frequency and the current diffusion timestep. We will modify the Introduction (Section 1.1) to make this motivation clear.
>
> **(2):** We also appreciate the reviewer’s point regarding **PCA eigenmode index**. We chose to use this proxy for analyzing the *training iterations* (unlike the diffusion timestep in Figure 1a) due to an empirical challenge: **in the early stages of training, the generated samples are heavily dominated by noise**. A direct 2D Fourier magnitude analysis of these early generated samples would be dominated by noise, masking the early structural patterns the model has just begun to learn. By following the framework of Wang & Pehlevan (2025) [1] and projecting samples onto a PCA basis pre-computed from clean training data, we effectively filter out the noise and focus on the emergence of meaningful data modes (detailed in Appendix D). We will add this explanation in the revised version.
>
> **W2: A second conceptual issue is that the actual learning ... what the optimization objective is actually doing.**
>
> **A:** We thank the reviewer for the detailed comment. In FaPS, the gating policy needs to decide which patch to select *before* the denoising loss is calculated. As defined in Section 3.1, its decision is conditioned on the input: the patch's frequency representation and the timestep. Therefore, the policy *must* rely on frequency information to evaluate the "**importance of each patch**" at the current stage. Without this frequency awareness, the policy would be making blind guesses. Thus, "frequency-aware patch selection"  accurately describes the feature space and mechanism driving the selection process.  In addition, we appreciate the reviewer’s comment. We clarify that our mechanism can be viewed as a form of adaptive curriculum learning. Specifically, both along the timestep and throughout the training process, the model tends to focus on low-frequency information first and gradually shift toward high-frequency details later.
>
> [1] Wang, B., & Pehlevan, C. An Analytical Theory of Spectral Bias in the Learning Dynamics of Diffusion Models. NeurIPS 2025

---

> > ### Author Rebuttal · Reviewer_okjv · 2026-04-05
> >
> > I'll keep my score as no further follow-ups are made.

---

> > > ### Author Response · Authors · 2026-04-07
> > >
> > > Thank you for taking the time to carefully read our rebuttal and acknowledge that some concerns have been partially addressed. We understand that the remaining concerns mainly relate to the conceptual framing and paper presentation rather than the algorithm itself. In the revision, we will strengthen the transition from the “dual spectral bias” observation to the design of FaPS, and clarify why the joint dependence on timestep and training iteration can support  to design a learnable adaptive policy.  We appreciate your feedback, which will help us improve the clarity and presentation of the paper.

---

### Decision · Program_Chairs · 2026-04-30

**Decision:**

Accept (regular)

**Comment:**

While reviewers agreed that the method has a reasonable level of novelty, the paper is well written and clearly organized, and FaPS achieves up to 3× faster training while maintaining comparable generation quality, two reviewers highlighted serious unresolved concerns about the paper's soundness and experimental validation during the discussion.

First, most importantly, the reported FID values are relatively high, failing to convincingly demonstrate final performance improvement. Second, the logical chain from the "dual spectral bias" observation to the proposed strategy is not fully convincing, and the convergence guarantee is relatively weak with a clear theory-practice mismatch. Third, the paper lacks sufficient ablation to confirm frequency awareness is truly necessary for the claimed gains, with key validations remaining uncompleted after the rebuttal.

In summary, given the remaining concerns raised by two reviewers, I recommend this submission for Weak Accept. I encourage the authors to revise and resubmit to an appropriate future venue.